



# Impacts of dynamic dust sources coupled with WRF-Chem 3.9.1 on the dust simulation over East Asia

Yu Chen[1], Yue Zhang[1], Siyu Chen[*1], Ben Yang[2], Huiping Yan[3], Jixiang Li[4], Chao Zhang[1], Gaotong Lou[1], Junyan Chen[1], Lulu Lian[1], and Chuwei Liu[1]

[1]Key Laboratory for Semi-Arid Climate Change of the Ministry of Education, Lanzhou University, Lanzhou 730000, China

[2]CMA-NJU Joint Laboratory for Climate Prediction Studies, School of Atmospheric Sciences, Nanjing University, Nanjing 210008, China

[3]School of Atmospheric Sciences, Nanjing University of Information Science and Technology, Nanjing 210008, China

[4]Key Laboratory of Land Surface Process and Climate Change in Cold and Arid Regions, Northwest Institute of Eco-Environment and Resources, Chinese Academy of Sciences, Lanzhou 730000, China

*Correspondence to: Siyu Chen (chensiyu@lzu.edu.cn)*

**Abstract:** Dust emission refers to the spatial displacement process of soil particles with the influence of wind. The quantitative and accurate description of dust emission is the basis of dust simulation in the modeling. The previous studies always employed static land cover in the numerical models, ignoring dynamic variations in the surface bareness and leading to large uncertainties in the dust simulation. We build six sets of dynamic dust sources functions, which shows a pronounced monthly and annual variability with the influence of seasonal change. Compared that in July, the dynamic dust source in March shows an expanding pattern to the edge of the deserts. Moreover, the dust source function in the Taklimakan Desert and Gobi Desert decrease at an annual rate of $2.42 \times 10^{-4}$ and $3.06 \times 10^{-4}$. The Weather Research and Forecasting model coupled to Chemistry (WRF-Chem) coupled with dynamic dust sources can effectively reproduce the spatiotemporal distribution of aerosol within satellite and ground-based observations. Our results show that the surface bareness and topographic characteristics jointly control the spatial distribution and value of dynamic dust sources. Further, the dynamic change of dust source further affects the dust emission and dust cycle. This study highlights the importance of surface bareness and the topographic characteristics on the dynamic dust source, and effectively improves dust cycle simulation over East Asia.

## 1. Introduction

The dust cycle is an important part of the Earth-atmosphere system (Wu et al., 2020). As one of the most abundant aerosols in the atmosphere, dust aerosols play a crucial role in the energy balance and hydrological cycle of the Earth system (Qian et al., 2011; Huang et al., 2010; Chen et al., 2018, 2022). Dust aerosols directly affect the energetic budget of the Earth-atmosphere system by scattering and





absorbing solar radiation (Sokolik et al., 2001; Balkanski et al., 2007; Zhao et al., 2010; Chen et al.,
2013), or they indirectly alter the radiation budget of clouds and the Earth by acting as cloud
condensation nuclei and ice nuclei to change the microphysical properties of clouds (Huang et al., 2006;
Kaufman et al., 1997). Moreover, dust deposition provides nutrients such as iron to the marine
ecosystem, changes the marine carbon dioxide budget, and regulates marine primary productivity by
promoting phytoplankton growth, thus affecting the marine biogeochemical cycle (Mahowald et al.,
2009). Dust aerosols are also easily enriched with acidic substances, bacteria, organic pollutants, and
heavy metals, which increases the number of inhalable particles in the atmosphere, thereby posing
serious threats to the air quality, human respiratory and cardiovascular systems (Zhao et al., 2008; Chen
et al., 2004; Thomson et al., 2006; Chen et al., 2019).
The improvement of dust modeling are crucial for improving the predictive accuracy of mesoscale
models and the accurate warning and prediction of dust weather (Gong et al., 2003; Uno et al., 2008;
Huang et al., 2010). Due to the complex dust involved physical processes, the quantity and properties
of dust simulated by numerical models differ greatly in different spatiotemporal scales. Huneeus et al.
(2011) systematically analyzed 15 global aerosol models included in the AeroCom plans (http://nansen.
Ipsl. Jussieu.fr/AEROCOM/), and they discovered substantial simulation differences in the dust
lifetime and dust climate effects. Generally, the simulated global average dust optical depth ranges
from 0.01 to 0.053, but the results of 80% of the models focus on 0.02–0.035. The simulation
differences in dust vertical integral parameters (such as AOD and column contents) between different
models are marginal, about 2 times. However, the simulated difference in the dust emission flux, total
deposition, and surface concentration is up to tenfold, and the simulated annual average dust emission
flux ranges from 500 to 4400 Tg, which is substantially larger than the estimated range (1000–2150 Tg)
in the climate dust model released by Zender et al. (2013). Additionally, the 15 models display
substantially different dust emission fluxes for Asia. The Goddard Chemistry Aerosol Radiation and
Transport (GOCART) simulation has a maximum value of 873 Tg, while the LSCE simulation has a
minimum value of 27 Tg. The difference between the two models is as large as 32 times, which is
much higher than the simulation differences worldwide, especially in North Africa and Central Asia.
The accuracy of dust emission simulation mainly depends on the spatial distribution of dust sources.
Correctly identifying the location of dust sources is a prerequisite for accurately simulating the dust
cycle in numerical models (Parajuli et al., 2019). However, the accurate identification of dust source
regions is very complicated because it is constrained by the heterogeneity of land covers, geological
environments, and soil chemical/physical characteristics (Parajuli et al., 2014). Most of the global dust
emissions are mainly concentrated in permanent deserts (Kim et al., 2017), which are regarded as dust
sources by current climate and weather models. Recently, the influence of human activities on land
cover and land use is becoming increasingly important because of the rapid development of agriculture
and urbanization. The current schemes also regard anthropogenic dust sources as climate-static surfaces
and seriously ignore the effect of dynamic changes in potential dust sources on dust emission (Ginoux
et al., 2001; Huneeus et al., 2011; Kim et al., 2013, 2017). Land use activities and land management
influence profoundly on dust emission (Webb et al., 2018, Xi et al., 2016). For example, dust emission
over East Asia mainly come from barren soil, accounting for 84% of the total dust emission, while



grasslands and croplands represent 15% and 7%, respectively (Wu et al., 2022). In the early twentieth
century, the continuous development of agriculture and the gradual expansion of farmland increased
dust loading by 500% in the western United States (Neff et al., 2008).
Vegetation conditions are closely associated with the dust emission level in dust source regions
(Engelstaedter et al., 2003). There is a significant statistical correlation between the Normalized
Difference Vegetation Index (NDVI) and dust loading in dust source regions (Zender and Kwon, 2005).
Time-varying vegetation data can effectively depict the dynamic changes in dust source regions and
improve the simulation of dust emissions (Tegen et al., 2002). Considering NDVI in dynamic dust
source, the time variation of dynamic dust source can be effectively reflected. To consider the dynamic
changes in land use and land cover in a numerical model, Kim et al. (2013) used the monthly average
NDVI to characterize the dynamic changes in potential dust sources in the GOCART dust emission
scheme for the first time. The researchers discovered that dust emission fluxes on farmland and sparse
grasslands have noticeable seasonal changes, with a maximum difference of 20%.
Based on the surface bareness map constructed using the NDVI, this paper considers multiple factors
for reasonably describing the spatial distribution of dynamic dust sources over East Asia, obtaining the
key dust emission factors for different land covers over East Asia, and improving the existing dust
emission schemes to improve dust emission over East Asia. The detailed organization of the paper is as
follows. Section 2 describes the construction of the surface bareness map and topographic feature
function dataset. The WRF-Chem model, GOCAT parameterization scheme, six sensitivity experiments,
and model evaluation data sets used in this study are introduced in detail. Section 3 presents the model
evaluation and uncertainty analyses. Section 4 contains the summary and discussion.
**2. Data and Methods**
**2.1  Construction of surface bareness map, terrain feature function, and dynamic dust source**
The dust source function (S) is determined by surface bareness (B) and topographic features (H) (Kim
et al., 2013). We firstly calculated the topographical depression features (H) using high spatial
resolution relative sea level altitude data with a horizontal grid number of 10800 (north–south direction)
× 21600 (east–west direction), then calculated the surface bareness using the MODIS NDVI data set.
Finally, the monthly global dynamic dust source function (S) between 2001 and 2020 was constructed
with these two datasets. However, the dynamic dust source function based on this calculation method is
not accurate and the regions with perennial ice and snow cover at high latitudes and urban surface also
showing dust source function maximum. Therefore, global snow cover data set and land cover data set
from MODIS observations are used to constrain the S. Next, the detailed calculation of B and H will be
carried out.
Dust sources used in previous GOCART simulations were based on average land covers from the
Advanced Very High Resolution Radiometer (AVHRR) satellite, which has no temporal variance
(DeFries and Townshend, 1994). Although the dust source constructed using this method matches well
the dust source observed by satellite, it is a static function that neither reflects land cover change nor
considers seasonal cycle of surface bareness. Therefore, a global-scale dynamic dust source has been

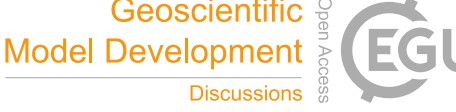

developed using time-varying NDVI data (Kim et al., 2013, 2017). The surface is bare where NDVI is
very low, while the ground vegetation cover increases with high NDVI. The corresponding equation is
as follows:
$\qquad B = N_{<thr}/N_{total}$,    (1)
where $N_{total}$ and $N_{<thr}$ indicate the total number of NDVI grid points and the number of grid points
where the NDVI value is less than thr for 0.5° × 0.5° grid cells, respectively. Additionally, thr is the
NDVI threshold, and the surface below the thr threshold is considered bare.
To consider the dust deposition accumulation from surface erosion in valleys and depressions (Ginoux
et al., 2001), the topographic feature H is defined as follows,
$\qquad H = \left(\dfrac{z_{max}-z_i}{z_{max}-z_{min}}\right)^5$,    (2)
where H represents the topographical depression features of each grid cell and the terrain elevation at
grid cell i, while $z_{max}$ and $z_{min}$ represent the topographic elevation of in the highest and lowest points
in the surrounding area, respectively. Notably, the spatial resolution of z is processed as 0.05° × 0.05°,
where the surrounding area refers to the the region of the grid point relative to the specific calculation
resolution (10° × 10°, 15° × 15°). The relative terrain height can be raised to the fifth power to increase
the terrain contrast.

### 2.2 WRF-Chem models

WRF-Chem model version 3.9.1 was employed in this study with Lambert projection and
unidirectional nested grids. The model area was centered at 36 °N and 105 °E, with a horizontal grid
number of 290 (east–west direction) × 240 (north–south direction) and a grid resolution of 20 km,
covering the whole of eastern China and the Gobi Desert and other major dust source regions.
Specifically, 36 layers extended from the surface to the model top at 100 hPa, with more layers in the
lower troposphere to better describe the boundary layer processes. The simulation period was selected
from February 27, 2020 to April 1, 2020. To avoid the influence of unstable simulation results caused
by initial conditions, only the results from March 1, 2020 to March 31, 2020 were considered in the
following analysis. Additionally, the anthropogenic emission inventory was obtained from the 2010
Global Atmospheric Research Emission Database-Hemispherical Transport of Air Pollution
(EDGAR-HTAP) global inventory with a horizontal resolution of 0.1° × 0.1°. EDGAR-HTAP provides
detailed inventory information on CH4, CO, SO2, NOx, NMVOCs, NH3, PM10, PM2.5, BC, and OC.
Moreover, biomass emissions based on the Model of Emissions of Gases and Aerosols from Nature
(MEGAN) were also selected. The parameterization schemes used in this study are shown in Table 1.
Table 1. WRF-Chem configuration options for physical and chemical parameterizations used in
this study

| | Physical and chemical processes | Configuration and reference |
| --- | --- | --- |
| | Microphysics | Thompson (Thompson et al., 2004) |
| Physical process | Long/shortwave radiation | RRTMG (Lacono et al., 2008) |
| | Land surface model | Noah (Chen and Dudhia, 2001) |





| | Boundary layer scheme | YSU (Hong et al., 2006) |
|---|---|---|
| | Cumulus parameterization | Grell–Devenyi (Grell et al., 2002) |
| | Dust emission estimation | GOCART (Ginoux et al., 2001) |
| Chemical process | Aerosol chemistry | MOZART Chemistry and GOCART |


### 2.3 Dust emission schemes


GOCART (Ginoux et al., 2001), including dust emission algorithms, transport, dry deposition and etc, have been added to the WRF-Chem model (LeGrand et al., 2019). As a relatively simple and highly empirical dust emission scheme, GOCART also has been widely welcomed by various numerical models and show excellent performance on dust emission over East Asia (Chen et al., 2014, 2017). Specifically, dust emission flux from GOCART is calculated as follows,

$$G = CSs_p u_{10m}^2 (u_{10m} - u_t), \quad u_{10m} > u_t, \quad (3)$$

where C ($\mu g\ m^{-2}\ s^{-1}$) is the constant of the dust emission factor, which is set to 1 $\mu g\ m^{-2}\ s^{-1}$. S is the dust source function based on the topography and surface parameters, and it is used to limit the dust emission area in the study area. $s_p$ represents the fraction of dust in each bin of particle size in the dust emission, where the particle size is represented by two lognormal distribution modes (accumulation mode and coarse mode). The median volume diameter and standard deviation for the accumulation mode is 2.91 ± 2.20 μm, while that for the coarse mode is 6.91 ± 1.73 μm. Additionally, $u_{10m}$ is the 10 m horizontal wind speed near the surface; $u_t$ indicates the threshold windspeed, which is a function of particle size, air density, and soil moisture.

### 2.4 Experiment design

To explore the impacts of surface bareness threshold and topographic depression feature calculation resolution, six different sensitivity experiments (DYN, DYN1, DYN2, DYN3, DYN4 and DYN5, see Table2) were designed to construct their impact for dynamic dust source over East Asia. In addition, one case (STA) using the original static dust source is also conducted and serves as a comparative experiment to verify the simulation effect of dynamic dust source on East Asia dust simulation. DYN was the dynamic dust source control experiment with a surface bareness threshold (thr) and topographic calculation grid resolution of 0.12 and 10° × 10°, respectively, and it was used as the standard for simulating dynamic dust sources in East Asia. Moreover, the difference between DYNx and DYN can be used to study the impacts of the surface bareness or the topographic characteristics on East Asian dust.


Table 2 WRF-Chem numerical experiments

| Cases | thr (surface bareness threshold) | topographic calculation grid resolution |
|---|---|---|
| STA | / | / |
| DYN | 0.12 | 10° × 10° |
| DYN 1 | 0.15 | 10° × 10° |
| DYN 2 | 0.17 | 10° × 10° |
| DYN 3 | 0.12 | 15° × 15° |




| | | |
|---|---|---|
| DYN 4 | 0.15 | 15° × 15° |
| DYN 5 | 0.17 | 15° × 15° |


## 2.5 Model evaluation data

Model evaluation was conducted using three datasets in this study. MODIS, an important sensor on
Terra, provides reliable global information on clouds, aerosols, and land covers. AERONET, which
uses the CIMEL automatic solar photometer (SPAM) as its basic instrument, is a ground-based aerosol
remote sensing network established by NASA and LOA-PHOTONS (CNRS). The network currently
covers major global regions and more than 500 sites. AERONET also plays an important role in
studying global aerosol transport, aerosol radiative effects, radiation transport patterns, and aerosol
results from satellite remote sensing. Three AERONET sites (Dalanzadgad (43.577 °N, 104.419 °E),
AOE_Baotou (40.852 °N,109.629 °E), and Beijing_RADI (40.005 °N, 116.379 °E)), which are very
close to dust sources, were selected here to explore the simulation effects of dynamic dust sources
while avoiding anthropogenic aerosols. In order to make the verification of dust simulation more
multi-source, UV aerosol index (AI) are employed in model evaluation, which is provided by Aura
OMI with a horizontal resolution of 1° × 1°. There are well-documented evidence for the connection of
UV AI and aerosol concentration and optical properties (Herman et al., 1997; de Graaf et al., 2005).
Moreover, AI is extremely sensitive to ultraviolet (UV)-absorbing aerosols, such as smoke, mineral
dust, and volcanic ash (Torres et al. 1998, Guan et al., 2010), which has a unique advantage in
simulating the spatial distribution of aerosols.
**3. Results**
**3.1 Global perspectives on surface erosion and topographic characteristics**
The greater the surface bareness, the drier the ground in arid and semi-arid regions, and the more likely
the dust is to be uplifted (Kim et al., 2013). Compared to the NDVI, surface bareness can better reflect
seasonal variations in soil bareness, thereby revealing detailed information on dynamic dust sources.
The thr threshold was selected as 0.12, 0.15, and 0.17 to determine the global perspective on surface
erosion in March 2020 (Fig. 1). The results revealed that when thr = 0.12, the global deserts and
high-latitude snow cover areas with a low NDVI were characterized by large surface bareness (B),
which could reach more than 0.9. Generally, B is small in rich-vegetation regions (such as eastern
China, India, most of South America, south-central Africa, and Indonesia), and the surface bareness can
be as low as 0. As the thr threshold increases, the surface bareness changes weakly in the center of the
global deserts and increases significantly at the edge of these deserts (Fig. 1b, c). When the thr
threshold was set at 0.17, the surface bareness in the southern margin of the Sahara Desert and northern
Central Asia could be increased by 0.7. Furthermore, the increased thr threshold also caused a slight
increase in surface bareness in Eurasia and Northern America. The surface bareness in Australia is
greatly affected by the thr threshold in terms of spatial range and numerical values.

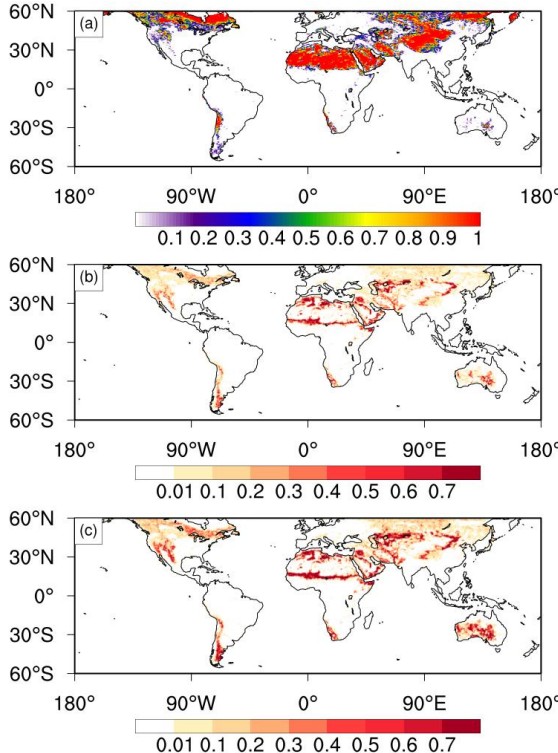

**Figure 1 Spatial distribution of the global surface bareness in March 2020. (a) thr is characterized by 0.12. (b) Surface bareness difference between thr = 0.15 and thr = 0.12. (c) Surface bareness difference between thr = 0.17 and thr = 0.12.**

Topographical depression features determine the relative height in the selected grid. The larger the topographical depression features, the more low-lying they are relative to the surrounding grids with more dust accumulation. Topographical depression features are large in typical permanent deserts, such as the Sahara Desert, the Australian Desert center, the Turkestan Desert and its northern part, the northern part of India, the Taklimakan Desert, and the Gobi Desert of the Mongolian Plateau, and they are always located in regions with a high probability of dust accumulation (Fig. 2). It was discovered that the topographic characteristics calculation resolution (10° × 10° and 15° × 15°) has a strong influence on topographic features. The increase in the grid resolution decreases topographic features in the center of the Australian Desert, the Taklimakan Desert, and the Gobi Desert of the Mongolian Plateau, whereas it slightly increases topographic features in other dust sources.



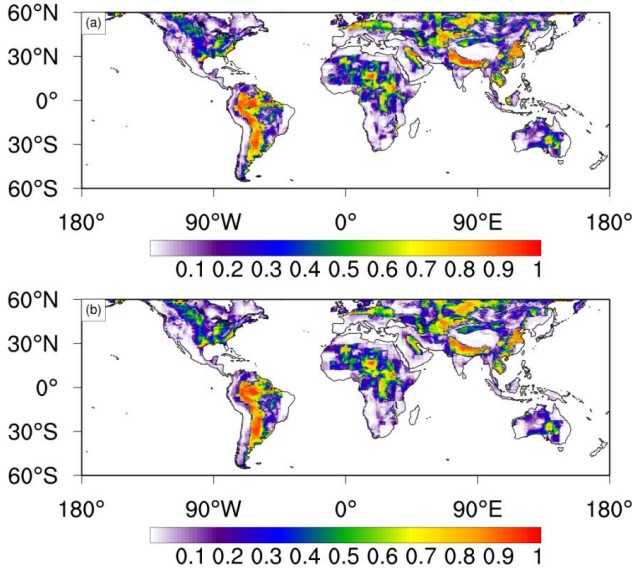

**Figure 2 Spatial distribution of global topographic features. (a) Calculation resolution of 10° × 10°. (b) Calculation resolution of 15° × 15°.**

Global dynamic dust sources for the recent 20 years (2001-2020) under six cases are constructed in this study. East Asia is an important dust source of the Earth (Wu et al., 2020). We further demonstrate the dynamic change of dust source function over the East Asia in the recent years. Results show that dynamic dust sources have pronounced fluctuations in different periods (Fig. 3). Specifically, dust eruption occurred frequently in spring over East Asia (Chen et al., 2023), the dust source function of the two deserts are generally larger than 0.3 in March (Fig. 3a). Compared that in July, the dust source function in March is also larger and expand to the edge of the desert (Fig. 3b). Exuberant vegetation is accompanied with low-bareness surface, and the dust source function in July is lower than that in March. The dust source function difference over the Taklimakan Desert and Gobi Desert also peak at 0.21 and 0.19 (Fig. 3b), respectively, which indirectly indicates the seasonal change impact great on the dust source function over East Asia. Moreover, the monthly variation of dust source function reaches the trough value in summer in different cases (Fig. 3c, d). After January and February, the dust source function decrease in March, April and May, which is related with the unfavorable growth of vegetation and large surface bareness in winter. The dynamic dust source function also shows sufficient annual variation characteristics (Fig. 3e, f).

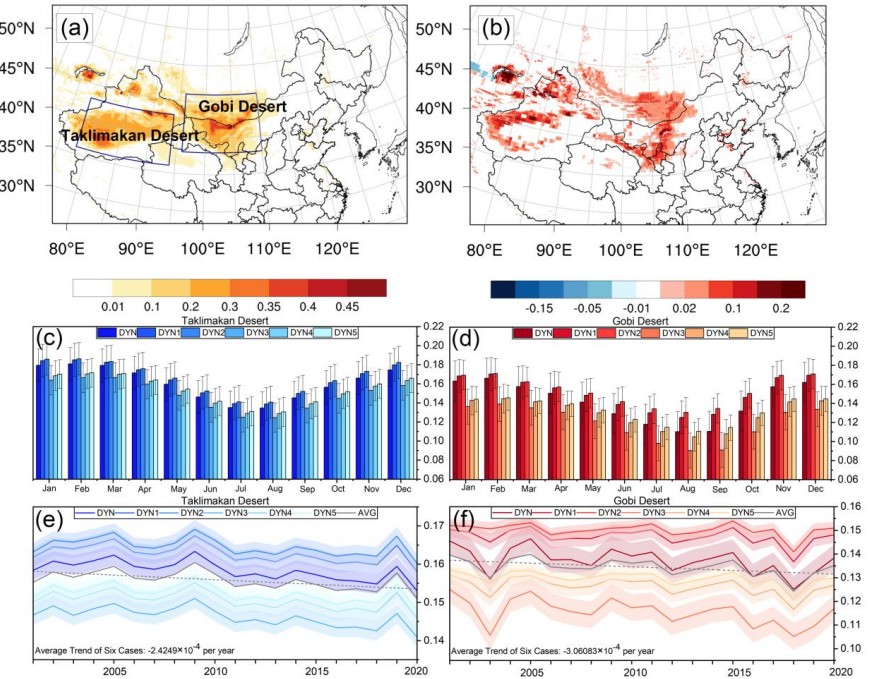

**Figure 3 Spatial distribution of averaged dust source function in the control experiment (DYN) in (a) March, and the (b) difference of dust source function between March and July from 2001 to 2020. The blue boxes indicate the Taklimakan Desert and the Gobi Desert. Monthly averaged dust source function in different cases from 2001 to 2020 in (c) Taklimakan Desert (36 °N–43 °N and 78 °E–94 °E) and (d) Gobi Desert (38 °N–46 °N and 96 °E–110 °E). Annual variation of dust source function in different cases in (e) Taklimakan Desert and (f) Gobi Desert; shading indicates one standard deviations from the 2001 to 2020 mean.**

As an important permanent desert over the East Asia, the dust source function of the Taklimakan Desert is larger than that of the Gobi Desert (Fig. 3c, d). Specifically, surface bareness and topographic characteristics calculation resolution impact greater on dust source function over the Gobi Desert than that over the Taklimakan Desert. The annual variation range of the dust source function in the Taklimakan Desert is 0.14~0.17, while that over the Gobi Desert is wider (0.1-0.15). The dust source function over the two deserts enhance with the surface bareness threshold. When the topographic characteristics calculation resolution increase to 15° × 15°, the fluctuations of dust source function over the Taklimakan Desert is around 0.012, while that in the Gobi Desert is 0.022. In addition, due to the climatic factors and the implementation of afforestation policy in China in recent years (Wu et al., 2022; Wang et al., 2023), the dust occurrence frequency has decreased, and the dust source function value





also show a downward trend. It decrease at a rate of $2.4249 \times 10^{-4}$ per year over the Taklimakan Desert
and $3.0608 \times 10^{-4}$ per year over the Gobi Desert.

**3.2  Uncertainty analysis of dynamic dust sources over East Asia**

Dust activity over the East Asia occurs frequently in spring (Wu et al., 2022). Therefore, we took
March 2020 as an example to deeply explore the impact of dynamic dust sources on the dust simulation
over the East Asia. Compared to that of static dust sources, the spatial distribution of dynamic dust
sources expands significantly when various land surfaces are treated as potential dust sources, and new
dust sources appear in both southern Mongolia and the Gobi Desert (Fig. 4STA, DYN). The
WRF-Chem coupled with dynamic dust source captures large dust source values in the Gobi Desert and
Taklimakan Desert, and the dust source value difference between cases DYN and STA reached 0.2 (Fig.
4DYN-STA). Due to the high dust emission rate and the large change of vegetation coverage in
southern Mongolia and central northern China, vegetation has great potential to the influence on dust
sources (Mao et al., 2013). As the surface bareness threshold increased, the dust source function
basically exhibited no change in the DYN regions, while new dust sources appeared on the grasslands
and farmland in the east and northwest of DYN. Additionally, the dust source function changed by
more than 0.05 over East Asia and could exceed 0.2 in Central Asia (Fig. 4a, b). There were even dust
sources in the Beijing–Tianjin–Hebei region and northeast China when the surface bareness threshold
was set as 0.17.
The topographic feature function calculated with different calculation resolutions is also crucial
for determining the dynamic dust source (Fig. 4c). The Tibetan Plateau intensifies the complexity of the
topography of East Asia. Therefore, changes in topographic features affect significantly on the two
major dust sources over East Asia. Unlike surface bareness, topographic features can affect dynamic
dust source values, but they have minor effects on the spatial distribution of the dynamic dust source
function. The coarse topographic characteristics calculation resolution inhibited the dynamic dust
source in the western part of the Taklimakan Desert and the southern part of the China–Mongolian
border, but it increased in the eastern part of the Taklimakan Desert and the northern part of the
China–Mongolian border (Fig. 4c). The cooperation between surface bareness and topographic
characteristics caused variations in the dynamic dust source for both the numerical size and spatial
distribution (Fig. 4d,e). It was further revealed that topographic features mainly affect dynamic dust
sources in the central dust region of East Asia, while surface bareness controls the development of
dynamic dust sources at the edge of these regions.

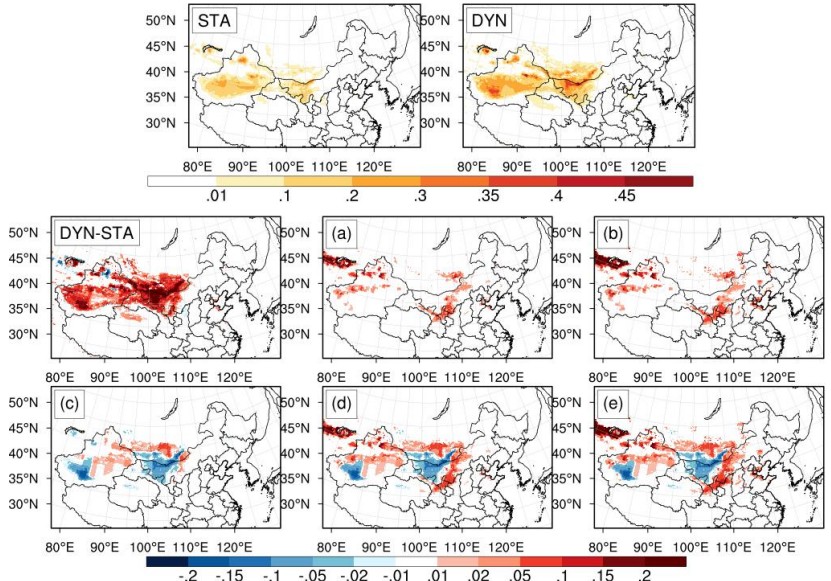

**Figure 4 Spatial distribution of monthly mean dust sources function from simulations in the STA and DYN cases in March 2020 and the difference between the DYN case and the DYN1, DYN2, DYN3, DYN4, and DYN5 cases: (a) DYN1–DYN, (b) DYN2–DYN, (c) DYN3–DYN, (d) DYN4–DYN, and (e) DYN5–DYN.**

### 3.3 Model evaluation

MISR is a reliable sensor for retrieving AOD in deserts (Christopher et al., 2008), and there is a high correlation between MISR AOD and AERONET AOD with its excellent observation and spectral capabilities (Cheng et al., 2012, Bibi et al., 2015). Cloud makes MISR AOD data gaps, which is a common phenomenon. However, the WRF-Chem coupled with dynamic dust sources effectively improves aerosol simulations in the dust sources by comparing satellite remote sensing data and the numerical model data. Compared with the simulated AOD in case STA, the dynamic dust source function changes dust emission, and the simulated AOD in the dust source regions also improved. Interestingly, by changing the surface bareness threshold and topographic characteristics calculation resolution, it was discovered that AOD variations are basically consistent with the dynamic dust source function. The increase in surface bareness was always accompanied by an increase in the AOD (0.01–0.04) in northwest China (Fig. 5a, b). However, the terrain changes weakened the AOD in northern Gobi Desert when compared to those in case DYN (Fig. 5c). Although an increase in the topographic characteristics calculation resolution causes the southern and northern parts of the Gobi Desert to exhibit opposite variations in dust sources, the southern part is more negatively affected by the topographic change in AOD.

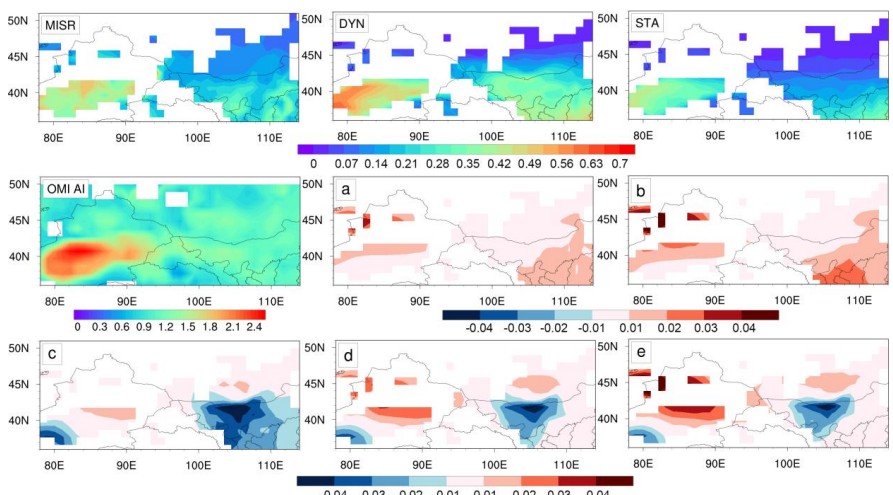

**Figure 5 Spatial distribution of the AOD from the MISR retrievals, the corresponding simulations for cases DYN and STA, OMI AI and the difference between the DYN case and the DYN1, DYN2, DYN3, DYN4, and DYN5 cases: (a) DYN1–DYN, (b) DYN2–DYN, (c) DYN3–DYN, (d) DYN4–DYN, and (e) DYN5–DYN.**

AI index is an indicator of the presence of aerosols in the atmosphere (Al-Zuhairi et al., 2021), which is often used to simulate the spatial distribution of aerosols. The simulated AOD in case DYN effectively shows the similar spatial distribution of aerosol with OMI AI. The WRF-Chem coupled with dynamic dust sources improved aerosol simulation in the Taklimakan Desert, the Gobi Desert, and northwest China (Fig. 5). Generally, the uncertainty of AERONET retrievals is less than that of MISR retrievals (Petrenko and Ichoku, 2013). The results revealed that AOD simulations in case STA were seriously underestimated when compared to the ground observations, whereas the AOD simulation in different dynamic cases were more consistent with the ground observations (Fig. 5). Using Dalanzadgad as an example, the difference in AOD between cases DYN and STA in the study period could reach 0.2, while the maximum AOD difference at Baotou peaked at 0.4. The correlation coefficient of the model and observations was as high as 0.8 in the three sites. It was even close to 0.9 at the Beijing station in cases DYN3 and DYN4.

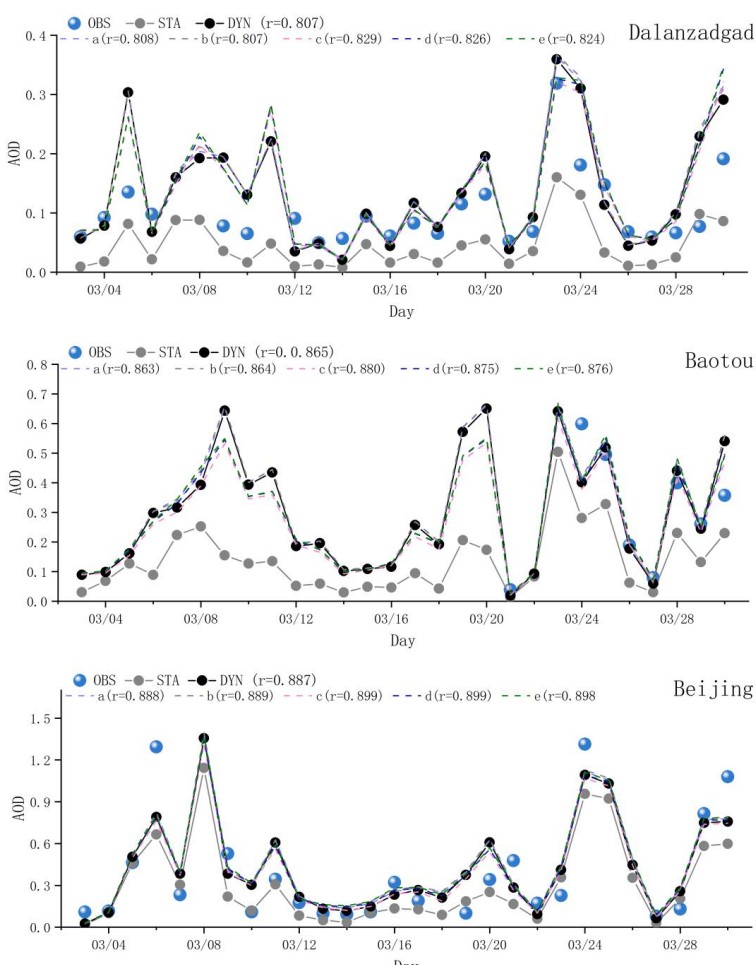

324

**Figure 6 Daily variations of AOD at 550 nm from AERONET observations (OBS) and the WRF-Chem model in different cases (DYN, STA, DYN1 (a), DYN2 (b), DYN3 (c), DYN4 (d), DYN5 (e)) during the same simulation periods at three sites (Dalanzadgad (43.577 °N,104.419°E), AOE_Baotou (40.852 °N,109.629 °E), Beijing_RADI (40.005 °N, 116.379 °E)).**

### 3.4 Uncertainty analysis of different dynamic dust source functions for the dust cycle

The regional average performance is illustrated in Fig. 7 to provide a better understanding of the influence of dynamic dust sources on the two major deserts over East Asia. Generally, the increase in surface bareness threshold caused the dust cycle to exhibit the largest physical quantities in case DYN2 (regional average dust emission flux: 0.9 μg m$^{-2}$ s$^{-1}$, dust loading: 0.2 g m$^{-2}$, and dust deposition flux: 79.7 μg m$^{-2}$ s$^{-1}$). When only the calculation resolution of the terrain feature was increased, the dust cycle presented the lowest value (regional average dust emission flux: 0.8 μg m$^{-2}$ s$^{-1}$, dust loading: 0.1 g m$^{-2}$, and dust-deposition flux: 68.9 μg m$^{-2}$ s$^{-1}$). Compared to the case of the control experiment (DYN), the combination of surface bareness and terrain features in DYN4 and DYN5 synergistically increased the dust cycle.





The Taklimakan Desert and the Gobi Desert are the two main dust sources over East Asia. The
influence of dynamic dust sources on the dust cycle in these two regions in different cases is consistent
with that over the study area. Dynamic dust sources have a more significant effect on the dust emission
and dust cycle in the Gobi Desert than on those in the Taklimakan Desert. The dust emission flux in the
Taklimakan Desert (average value: 3.8 μg m$^{-2}$ s$^{-1}$) shows a weak change in different cases. Although
the overall dust emission flux in the Taklimakan Desert is about 1.69 times less than that of the Gobi
Desert, the deposition flux in the Taklimakan Desert (~350 μg m$^{-2}$ s$^{-1}$) is equal to that in the Gobi
Desert. The dust loading in the Taklimankan Desert (average value of the six experiments is 0.62 g m$^{-2}$)
is even ~1.4 times larger than that in thet Gobi Desert (mean: 0.44 g m$^{-2}$ for the six experimental
groups). The dynamic dust sources in the Gobi Desert have a particularly significant influence. In the
six experiments, the maximum difference in dust emission in the Gobi Desert was up to 1 μg m$^{-2}$ s$^{-1}$,
which is much larger than the average value in the study area.

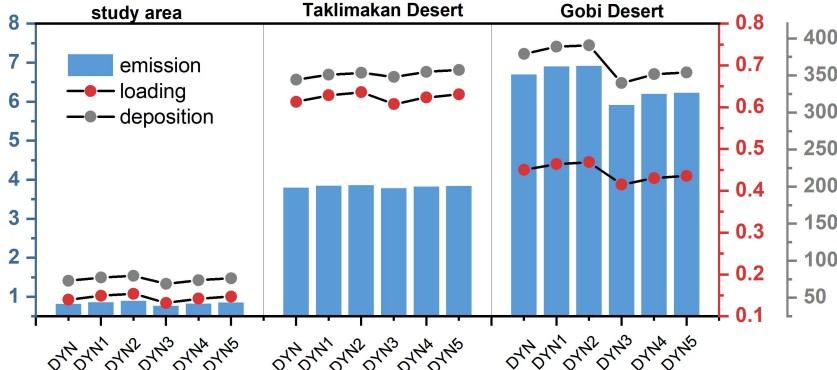


**Figure 7 Regional average of the dust emission flux (blue bar graph, units: μg m$^{-2}$ s$^{-1}$), dust loading (red dots, units: g m$^{-2}$), and dust deposition flux (gray dots, units: μg m$^{-2}$ s$^{-1}$) in the study area (13 °N–51 °N and 78 °E–127 °E), Taklimakan Desert (36 °N–43 °N and 78 °E–94 °E), and Gobi Desert (38 °N–46 °N and 96 °E–110 °E) in the different cases (DYN. DYN1, DYN2, DYN3, DYN4, and DYN5).**

Dust emission is an important uncertainty factor in dust cycle simulation, and it is closely associated
with surface composition, land use, and soil moisture (Yahi et al., 2013). Dust cycle parameters are the
most intuitive indexes for the construction effect of dynamic dust sources. Therefore, the simulation
effects of dynamic dust sources on dust cycle parameters are discussed based on six dynamic dust
source functions constructed using two parameters (surface bareness and topographic characteristics).
Dust emission is mainly concentrated in the Taklimakan Desert and Gobi Desert (Fig. 8a), with
maximum dust emission flux peaks at 50 μg m$^{-2}$ s$^{-1}$. The dust source and dust emission show a
coherent distribution in North China. Notably, there are small amounts of dust emissions in the western
Tibetan Plateau (Fig. 8b, c). Surface bareness is important to the spatial distribution of dust emissions
in dust sources. Grasslands and farmland are treated as potential dust sources with a large surface
bareness. Therefore, as the surface bareness threshold increases, more dust emissions will appear in the
Taklimakan Desert, the eastern and northwestern parts of the Gobi Desert, and the eastern part of the
DYN Tibetan Plateau (Fig. 8b, c).
Topographic features considerably affect dust emissions by changing the dynamic dust source. Figure
8d shows the influence of topographic characteristics functions on the dynamic dust emission under
different calculation resolutions. The topographic characteristics of the coarse calculation resolution
had an inhibitory effect on the dynamic dust emission simulation in the western Taklimakan Desert and
the southern part of the China–Mongolia boundary. However, in case DYN3, dust emissions increased
in the eastern part of the Taklimakan Desert and the northern part of the China–Mongolia boundary.
Moreover, owing to the combined effects of surface bareness and topographic characteristics on the
dynamic dust sources, the dust emission flux increased in grasslands and farmland in the east and
northwest of DYN areas and decreased in the western Taklimakan Desert and the southern part of the
China–Mongolia border (Fig. 8e f). Therefore, the combination of these two factors will inhibit dust
emission in the central Gobi Desert while promoting dust emission in its marginal areas. Additionally,
it will suppress and promote dust emissions in the western and eastern parts of the Taklimakan Desert,
respectively.

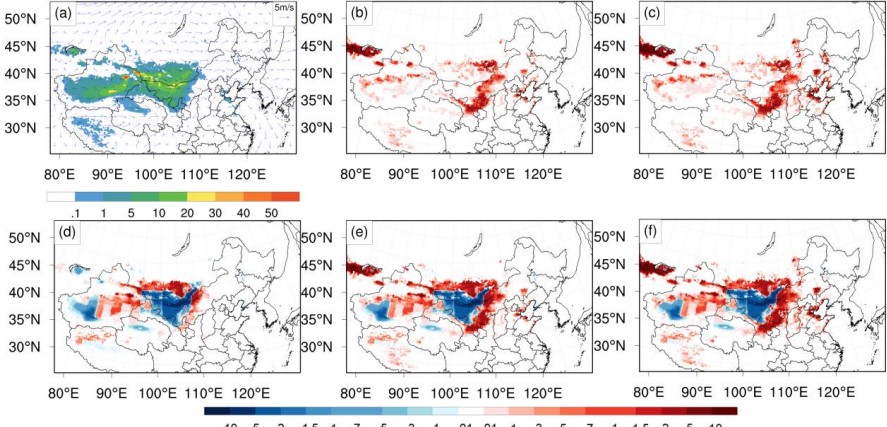

**Figure 8 Spatial distribution of dust emission (color contour, units: μg m$^{-2}$ s$^{-1}$) from the WRF-Chem**
**simulations in (a) case DYN (vector, units: m s$^{-1}$) and the difference between the DYN case and the DYN1,**
**DYN2, DYN3, DYN4, and DYN5 cases: (b) DYN1–DYN, (c) DYN2–DYN, (d) DYN3–DYN, (e) DYN4–DYN,**
**and (f) DYN5–DYN.**
The six dynamic dust sources also caused a significant difference in dust loading and dust dry
deposition. The Taklimakan Desert had a large dust loading, with a maximum peak of 1.5 g m$^{-2}$. Dust
loading in the Taklimakan Desert was much larger than that in the Gobi Desert, which is consistent
with regional averages (Fig. 7). Additionally, it could be transported eastward to South Korea and
Japan as well as southward to most provinces in southern China and the Tibetan Plateau. Notably, as
the surface bareness threshold increased, the change in dust loading in the DYN cases remained
consistent with the spatial distribution of the dust emission flux (Fig. 9b, c). Both cases DYN1 and
DYN2 simulated large dust loading in central and northern China. It is noteworthy that the relative
difference in dust loading between cases DYN, DYN1, and DYN2 was significantly lower than that in
the dust emission flux in northern China. Moreover, the increase in the topographic characteristics
calculation resolution also decreased the dust loading in North China and in the middle and lower parts
of the Yangtze River Basin (Fig. 9d). Unlike the dust emission, larger topographic calculation
resolution only increase dust loading in eastern Taklimakan and northern Gobi Desert (Fig. 9e), while





larger dust emission appear in the eastern part of Gobi Desert (Fig. 8e). It is indicated that the source
area of the Gobi desert has a greater influence on dust loading.

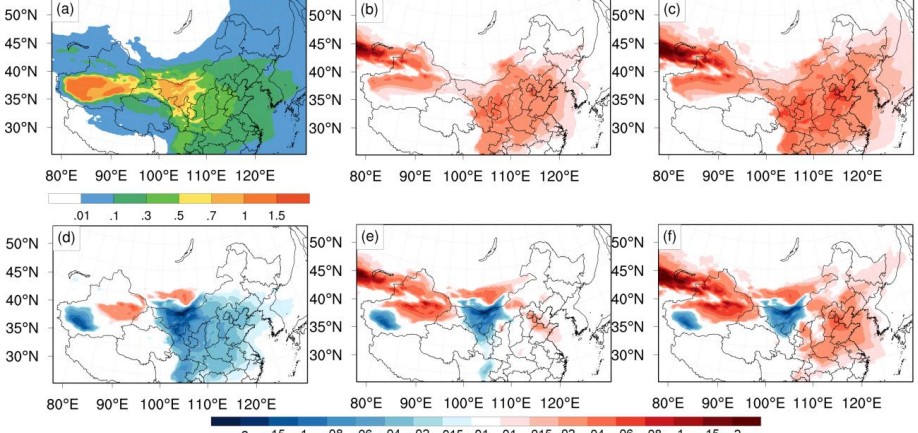

**Figure 9 The same as Figure 8 but for dust loading (units: g m⁻²).**
Figure 10 shows the vertical profile structure of the dust concentration at 40 °N to provide a further
understanding of the influence of the dynamic dust sources on the dust vertical structure. The dust
concentration was mainly high at 80 °E–100 °E in case DYN (Fig. 10a), and the maximum value was
over 500 μg m⁻³. The vertical dust transmission could reach about 9 km, and it was more than 50 μg
m⁻³ below 6 km. The dust concentration gradually decreased at 125 °E. As the surface bareness
threshold increased (Fig. 10b, c), the area with a large difference in the simulated dust concentration
expanded eastward significantly, reaching up to 130 °E and up to 5 km vertically, with the maximum
difference exceeding 30 μg m⁻³. The simulated dynamic dust concentration calculated using the large
topographic characteristics calculation resolution increased by more than 20 μg m⁻³ at 80 °E–95 °E (Fig.
10d), reaching 4.5 km vertically. The dust concentration in other longitude areas decreased by more
than 30 μg m⁻³, reaching about 7 km vertically. Figure 10e and f show that surface bareness and terrain
features jointly influence the vertical dust concentration. Interestingly, the impact of surface bareness
on dust concentration is greater than that of topographic features east of 113 °E, and the area with a
large difference in dust concentration extends eastward to 130 °E, reaching 4 km vertically.

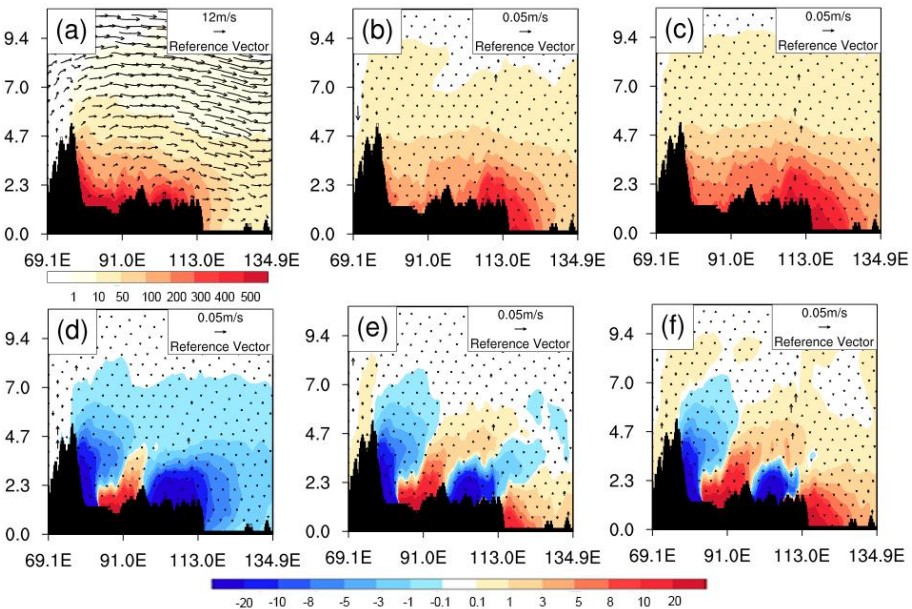

**Figure 10 Profile of dust concentration at 40° N in the different DYN cases (color contour, units: μg m⁻³; vector, units: m s⁻¹). (a) Dust concentration profile for case DYN. (b), (c), (d), (e), and (f) represent the difference between the DYN case and the DY1, DYN2, DYN3, DYN4, and DYN5 cases (DYN1–DYN, DYN2–DYN, DYN3–DYN, DYN4–DYN, and DYN5–DYN), respectively.**

These cases simulate different dust dry depositions according to their different dust sources and dust size distributions. Dust emission is closely related to soil texture, soil water content, atmospheric stability and near-surface wind speed (Marticorena & Bergametti, 1995). It also mainly concentrate on dust source regions (Fig. 8). Dust emission is the main factor determining the atmospheric dust concentration, while dust dry deposition depends on dust concentration and dust dry deposition velocity. The relationship between dust emission and dust deposition is not completely linear. Therefore, the spatial distribution of dust emission and dust dry deposition are similar with each other, but not completely consistent. The maximum dust dry deposition flux was greater than 1000 g day⁻¹ in case DYN (Fig. 11a). Dust dry deposition flux also has more heterogeneous than that of wet deposition flux with less precipitation over the deserts (Hu et al., 2019). Dust dry deposition fluxes are closely associated with dust mass loading, while dust wet deposition fluxes are determined by both precipitation and mass loading (Zhao et al., 2013).Therefore, the spatial distribution of dust dry deposition was similar to that of dust loading (Fig. 9). Generally, cases DYN1 and DYN2 with larger surface bareness displayed more dust dry deposition in Central Asia, the Taklimakan Desert, and North China (Fig. 11b, c). Similar to the spatial distribution of dust loading, the increase in the topographic feature calculation resolution also weakens dust deposition in the Gobi Desert and the western part of the Taklimakan Desert.

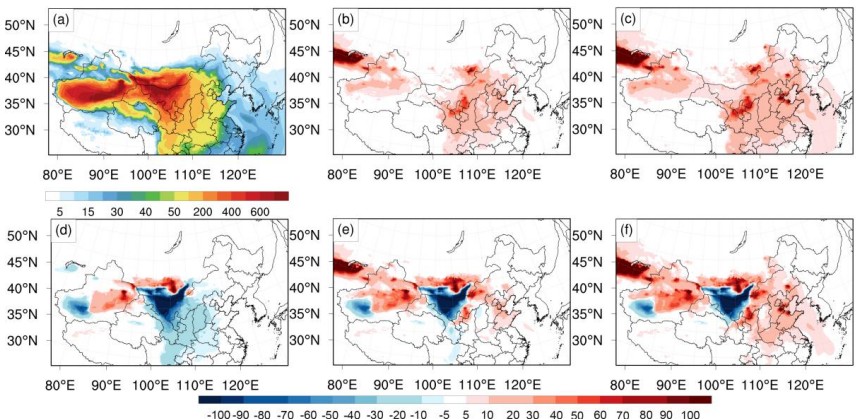

**Figure 11 The same as Figure 8 but for the dust dry deposition flux (color contour, units: g day$^{-1}$).**
Dust aerosols over East Asia are characterized by their high concentrations and absorbability (Chen et
al., 2022). Dust aerosols can further affect weather and climate changes over East Asia by absorbing
solar shortwave radiation and changing the energy budget of the Earth-atmosphere system. Radiative
forcing is also an important index for evaluating the impact of aerosols on climate change. Chen et al.
(2014) stated that the average dust direct radiative forcing over East Asia at the top of the atmosphere is
about −2.0 W m$^{-2}$ based on the WRF-Chem. The spatial distribution of dust radiative forcing under the
above mentioned six experiments is shown in Fig. 12 to further explore the uncertainty of different
dynamic dust sources on dust radiative forcing over East Asia. The net radiative forcing of dust at the
top of the atmosphere is mainly negative in the main dust sources and downstream regions, such as
East China, with a maximum of −16.7 W m$^{-2}$, indicating that dust has a significant cooling effect on
the ground-atmosphere system.
Surface albedo is one of the most important factors affecting dust radiative forcing (Ma et al., 2012).
Therefore, as surface bareness threshold increases, the spatial distribution of net dust radiative forcing
in Central Asia changes slightly, but the magnitude increases. Contrary to the variation in basic dust
cycle parameters, the influence of dynamic dust sources on dust radiative forcing is mainly
concentrated in central China and North China. The influence of dynamic dust sources on dust radiative
forcing at the top of the atmosphere could be extended to downstream regions, such as Japan and South
Korea, when the surface bareness threshold is set to 0.15. Moreover, as the topographic characteristics
calculation resolution increases, the negative dust radiative forcing at the top of the atmosphere
decreases in northwest China but slightly increases in Mongolia. The combination of topographic
characteristics and surface bareness causes opposite changes in the radiative forcing in the Gobi Desert
in China and Mongolia when compared to the spatial distribution of dust loading (Fig. 9e, f), further
clarifying that dust has a significant cooling effect on the top of the atmosphere.

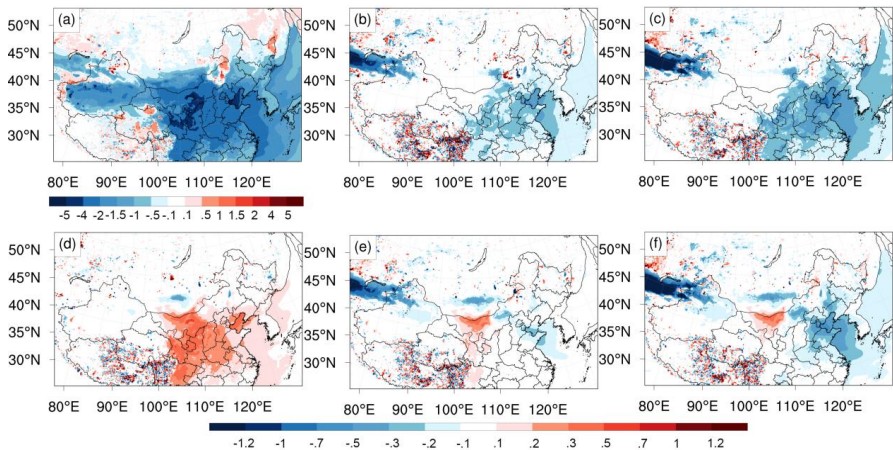


**Figure 12 The same as Figure 8 but for dust radiative forcing at the top of the atmosphere (units: W m$^{-2}$).**

### 4. Discussions and Conclusion

The dynamic dust source constructed in this study uses various land surfaces as potential dust sources. Compared to that coupled with static dust sources, the WRF-Chem coupled with dynamic dust sources can effectively reduce the uncertainty of dust emission simulation, which is important for preventing and controlling wind dust and desertification over East Asia, as well as understanding the impact of land use changes on air pollution in the future. Six dynamic dust sources were constructed based on different surface bareness (NDVI thresholds: 0.12, 0.15, and 0.17) and topographic features (calculation resolutions: $10° × 10°$ and $15° × 15°$), and their influence on dynamic dust sources is also revealed herein. The constructed dynamic dust source function has a pronounced temporal variability. Compared that in July, the dust source over the Gobi Desert and Taklimakan Desert expand to the edge in March, which is connected with more vigorous vegetation growth in summer. Moreover, the dust source function also shows an obvious monthly and annual variation. However, Taklimakan Desert is a typical permanent desert over the East Asia. The dynamic dust source change over the Taklimakan Desert is smaller than that over the Gobi Desert. The dust source function of the Taklimakan Desert and Gobi Desert also decrease at an annual rate of $2.42 × 10^{-4}$ and $3.06 × 10^{-4}$. The spatial distribution of the dynamic dust source was significantly larger than that of the static dust source. New dust sources appeared in southern Mongolia and the Gobi Desert at the China–Mongolia border, with a dust source of >0.1. The WRF-Chem effectively improved dust simulation across the dust source regions when coupled with dynamic dust sources. The spatial distribution of the AOD simulation derived from the dynamic dust sources was consistent with that of MISR AOD, and effectively show the spatial distribution of aerosol observed by OMI. However, AOD simulations based on static dust sources were not in good agreement with those of MISR. Moreover, the correlation coefficient between AERNET AOD and the WRF-Chem AOD was even larger than 0.8.

This study also examines the uncertainties resulting from dynamic dust sources in dust cycle simulation over East Asia. Our results revealed that changes in surface bareness and topographic





characteristics could change basal parameters of dust cycle (dust AOD, dust emission flux, dust loading,
dust concentration at different height layers, and dust dry deposition) by influencing the dynamic dust
sources. Overall, surface bareness and topographic characteristics considerably affect the spatial
distribution and numerical value of the dust cycle. The dust cycle simulation in the different DYN
cases differed from each other, but changes in the value and spatial distribution were consistent with
the changes in the dynamic dust sources. The simulation of the dust cycle in the eastern Gobi Desert,
Taklimakan Desert, and North China increases as the surface bareness increases, but that in the western
Taklimakan Desert and southern Gobi Desert decreases as the topographic characteristics calculation
resolution increase.
Only the main factors that affect dynamic dust sources, such as terrain and vegetation cover, were
considered in this study. Using vegetation coverage as an example, the use of the NDVI can effectively
address the problem of the current numerical models only identifying the climatic dust source regions
and reflecting no seasonal variation characteristics of the land cover. However, as an important index of
vegetation change, the NDVI still has some uncertainties in describing land types. It is possible to
mistake dead plants for a bare surface by employing a specific NDVI as the surface bareness threshold
(Kim et al., 2013), thereby causing a deviation in the recognition of dynamic dust source areas.
Therefore, the NDVI cannot accurately capture changes in bare ground. To accurately describe the dust
cycle over East Asia, it is worthwhile to find more accurate land use indexes (such as particle size
distribution, soil texture, water content, and dust particle content) and integrate them into the numerical
models for better dust simulation across East Asia in future studies. Although the dynamic dust source
was coupled with the WRF-Chem model, it was still driven by the classical dust emission
parameterization schemes, resulting in a large deviation in dust simulation outside fixed deserts. Land
cover types over East Asia are complex and diverse, and their dust emission mechanisms vary greatly.
Therefore, the current understanding of the dust emission mechanism of different dust sources over
East Asia is relatively preliminary.
In the future, a dust observation network over East Asia should be established to obtain key factors,
such as dust friction velocity, particle spectrum distribution, soil surface roughness, and water content
on different surfaces. Wind tunnel tests should be conducted on different land cover types to investigate
critical factors. The dust emission characteristics of different land cover types should be revealed, and a
dust emission parameterization scheme suitable for each land cover type should be constructed and
coupled to the regional model for simulation evaluation.
**Code and data availability**
The model code of WRF-Chem 3.9.1 released on August 17, 2017, are available at
https://doi.org/10.5065/D6MK6B4K. The source function is calculated with surface bareness and
terrain feature. The surface bareness is depended on MODIS NDVI, which are provided on
https://lpdaac.usgs.gov/products/mod13c2v006/. The topographic characteristics are calculated on
topographic                     elevation                                    from



https://www.gebco.net/data_and_products/historical_data_sets/#gebco_one. The code availability
for the construction of dynamic dust source and the constructed dynamic dust source over the
recent 20 years (2001-2020) are available in the supplement. OMI AI are available at
https://disc.gsfc.nasa.gov/datasets/OMTO3d_003/summary. AOD derived from MODIS are provided
from          https://ladsweb.modaps.eosdis.nasa.gov/missions-and-measurements/products/MOD08_D3.
AERONET AOD is publicly available at https://aeronet.gsfc.nasa.gov/. FNL reanalysis data are
accessed       from      http://dss.ucar.edu/datasets/.      Cam-chem      data      were      used      from
https://www.acom.ucar.edu/cam-chem/cam-chem.shtml. The NCL codes used to run the analysis can
be obtained in the supplement.
**Author contributions**
Siyu Chen designed the study. Yue Zhang contructed the dynamic dust source. Yu Chen and Yue Zhang
conducted paper writing and data analysis. Yu Chen, Yue Zhang, Siyu Chen, Ben Yang, Huiping Yan,
Jixiang Li, Chao Zhang, Gaotong Lou, Junyan Chen, Lulu Lian, Chuwei Liu contributed to the
discussion and paper writing.
**Competing interests**
The authors declare that they have no conflicts of interest.
**Acknowledgements**
This work was jointly supported by the Project supported by the Joint Fund of the National Natural
Science Foundation of China and the China Meteorological Administration (No. U2242209), the
National Natural Science Foundation of China (Grant. No. 42175106).

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
