# Peer review of "Impacts of dynamic dust sources coupled with WRF-Chem 3.9.1 on the dust simulation over East Asia"

_Geoscientific Model Development, 2023_

## Author Comment (AC1)

We thank the reviewers for his/her helpful comments and suggestions on improving our manuscript. These comments are incorporated into the manuscript now. Below is our point-by-point response to these comments. The reviewers' comments are in italic font, our responses are in normal font and the revision in the manuscript are identified in blue font.

*Community Comments1:*
*Thank you for sharing your research.*
*P2 L61: "The accuracy of dust emission simulation mainly depends on the spatial distribution of dust sources" – do you have a reference for this statement? I'd suggest removing this comment otherwise. Dust emission simulation errors can emanate from several sources (e.g., erroneous wind or environmental forcing condition patterns, erroneous soil/terrain/vegetation characterization, poorly or overly tuned empirical constants, etc.).*

**Reply:** Thanks for your suggestions. Uncertainites of dust emission simulation can emanate from several sources including wind (Chen et al., 2014), environmental forcing condition patterns (Shao et al., 2013), soil texture(Tian et al., 2021), terrain (Prospero et al., 2002), vegetation characterization (Tegen et al., 2002), poorly or overly tuned empirical constants (Zhao et al., 2010), etc. However, due to the strong spatial heterogeneity of global dust source (Kok et al., 2021), the accurate description of dust sources in numerical models has also become an important factor affecting the dust simulation uncertainties (Kim et al., 2013).

Considering the accurate expression in the manuscript, we also revised it in Lines 44–59 as follows: "The improvement of dust modeling are crucial for improving the predictive accuracy of mesoscale models and the accurate warning and prediction of dust weather (Gong et al., 2003; Uno et al., 2008; Huang et al., 2010). Due to the complex dust involved physical processes, the quantity and properties of dust simulated by numerical models differ greatly in different spatiotemporal scales. Huneeus et al. (2011) systematically analyzed 15 global aerosol models included in the AeroCom plans (http://nansen.ipsl.jussieu.fr/AEROCOM/). They found that the 15 models display substantially different dust emission fluxes for Asia. The Goddard Chemistry Aerosol Radiation and Transport (GOCART) simulation has a maximum value of 873 Tg yr$^{-1}$, while the UIO CTM simulation has a minimum value of 27 Tg yr$^{-1}$. The difference between the two models is as large as 32 times, which is much higher than the simulation differences worldwide, especially in North Africa and Central Asia. Uncertainites of dust emission simulation can emanate from several sources including wind (Chen et al., 2014), environmental forcing condition patterns (Shao et al., 2013), soil (Tian et al., 2021), terrain (Prospero et al., 2002), vegetation characterization (Tegen et al., 2002, ), poorly or overly tuned empirical constants (Zhao et al., 2010), etc. However, due to the strong spatial heterogeneity of global dust source (Kok et al., 2021), the accurate description of dust sources in numerical models has also become an important factor affecting the dust simulation uncertainties (Kim et al., 2013)."

*2) Sect. 2.3: Which WRF-Chem dust emission module is this section describing? Based on the description provided, it appears the authors used the dust_opt = 13 WRF-Chem namelist option, which is an adapted version of the GOCART dust emission scheme coupled with the MADE/SORGAM and MOSAIC aerosol scheme. If so, please reference the model description and publication by Zhao et al. (2010). Usually, when WRF-Chem users reference the GOCART Ginoux et al. (2001) scheme (e.g., Table 1), they're referring to the dust_opt = 1 setting. If the authors added new dust emission equations to the WRF-Chem code (i.e., if the authors did not use dust_opt = 1, 2, 3, 4, or 13), please state that in the manuscript.*

*Please note that if the authors did use dust_opt = 13, the dust emission code for this study deviates from the original formulation described in the paper by Ginoux et al. (2001). These modifications are described in detail in Sect. 3.1.2 of LeGrand et al. (2019). While the LeGrand et al. (2019) overview was written specifically for dust_opt = 1, most of these same changes also apply to the dust_opt = 13 setting (see function sorgam_source_du in module_aerosols_sorgam.F in the chem directory of the WRF-Chem source code). The most notable change is the switch from a 10m wind speed-based threshold (ut) to one derived in terms of wind friction speed (u\*t). This modification leads to spurious dust emissions under very low wind speeds since u10m>> u\*t. The dust_opt = 13 setting from WRF-Chem v3.9.1 also includes a degree of saturation value (θs) threshold (gwet in the code) that restricts dust emissions anywhere θsexceeds 0.2 (i.e., very dry conditions). This aspect of the dust emission module may be important to the authors' conclusions given the emphasis on green vegetation with NDVI.*

*Effectively, the dust emission flux (G) in dust_opt = 13 more closely resembles the following:*

*G=CSsp(u10m)3 if θs< 0.2; G= 0, otherwise.*

**Reply:** We are sorry for the readers' misunderstanding. The original GOCART-WRF scheme (dust_opt=1) is selected in the namelist.input configuration file. We have also clarified our expression for specific information about the original GOCART dust emission scheme in Lines 160–183 as follows: "The current dust emission schemes could be divided into three categories: empirical dust emission scheme, simplified-physical-processed dust emission scheme and detailed-microphysical-process dust emission scheme. The GOCART dust emission scheme is a representative of empirical dust emission scheme, which mainly considers the statistical relationship between dust emission flux and friction velocity. It has been widely used in dust emission simulation. Dust uplifting to the atmosphere is parameterized in GOCART dust emission scheme assuming that the vertical particle flux is proportional to the horizontal wind flux and representing the direct conversion from wind speed to dust emission (Ginoux et al., 2001). The impact of saltation bombardment on mobilization are internalized in the relationship between wind speed and dust emission.

The GOCART dust emission scheme (dust_opt=1) in the WRF-Chem need necessary input factors including wind speed, soil moisture, air density and generalized soil characteristics. The erodible soil is consist of sand, silt and clay. Specifically, five

ranges of dust bins (bin1: 0.1-1.0 μm, bin2: 1.0-1.8 μm, bin3: 1.8-3.0 μm, bin4: 3.0-6.0 μm, bin5: 6.0-10.0 μm) are used in the GOCART aerosol module. Dust emission flux from GOCART is calculated as follows,

$$F = \begin{cases} CSs_p u_{10m}^2 (u_{10m} - u_t), & u_{10m} > u_t \\ 0, & otherwise \end{cases} \quad (3)$$

where C is the constant of the dust emission factor. S is the dust source function based on the topography and surface parameters, and it is used to limit the dust emission area in the study area. The layer 1 of EROD parameter, provided by the WRF-Chem, is used to parameterize the S in Eq. (1). $s_p$ represents the fraction of dust in each bin of particle size in the dust emission. The default $s_p$ from Eq. (3) are {0.1, 0.25, 0.25, 0.25, 0.25}. Additionally, $u_{10m}$ is the 10 m horizontal wind speed near the surface; $u_t$ indicates the threshold wind speed. It is calculated as

$$u_{*t} = 0.129 \frac{(\frac{\rho_g g D_p}{\rho_a})^{0.5}(1+\frac{0.006}{\rho_p g D_p^{2.5}})^{0.5}}{[1.928(a(D_p)^x+0.38)^{0.092}-1]^{0.5}} \quad (4)$$

where x equals 1.56, a equals 1331 cm$^{-x}$."

*3) This study used an older version of WRF-Chem. There was an important bug fix added to the WRF-Chem dust gravitational settling scheme in version 4.1 (see Ukhov et al. 2021 and https://github.com/wrf-model/WRF/commit/2ffdebf4ac311a5b1ef8cd0c639e0d857b 550fdb). The error causes dust to fall out of the atmosphere too quickly. While redoing the experiment with a newer model version may not be necessary, how might this error affect the interpretation of the results?*

**Reply:**Thanks for your valuable suggestions. According to your suggestions, we have updated the manuscript to a newer version (WRF-Chem V4.4.2). The AOD simulation effect over the Gobi Desert in WRF-Chem 4.4.2 is closer to that of MODIS AOD than WRF-Chem 3.9.1. Moreover, the AOD simulation in dynamic dust source case over the Taklimakan Desert is better than that with the static dust source. We also have updated the revised manuscript (the revised Figs. 1-12) according to WRF-Chem 4.4.2.

[Figure]

**Figure 3: Spatial distribution of averaged dust source function in the control experiment (DYN) in (a) March, and the (b) difference of dust source function between March and July from 2001 to 2021. The blue boxes indicate the Taklimakan Desert and the Gobi Desert. Monthly averaged dust source function in different cases from 2001 to 2021 in (c) Taklimakan Desert (36°N–43°N and 78°E–94°E) and (d) Gobi Desert (38°N–46°N and 96°E–110°E). Annual variation of dust source function in different cases in (e) Taklimakan Desert and (f) Gobi Desert; shading indicates one standard deviations from 2001 to 2021.**

[Figure]

**Figure 4: Spatial distribution of monthly mean dust sources function from simulations in the STA and DYN cases in March 2020 and the difference between the DYN case and the DYN1, DYN2, DYN3, DYN4, and DYN5 cases: (a) DYN1−DYN, (b) DYN2−DYN, (c) DYN3−DYN, (d) DYN4−DYN, and (e) DYN5−DYN.**

[Figure]

**Figure 5: Spatial distribution of the AOD in March 2020 from the (a) MODIS retrievals, the corresponding simulations for cases (b) DYN3 and (c) DYN, and the difference between the DYN case and the DYN1, DYN2, DYN3, DYN4, and DYN5 cases: (d) DYN1−DYN, (e) DYN2−DYN, (f) DYN3−DYN, (g) DYN4−DYN, and (h) DYN5−DYN.**

[Figure]

**Figure 6: Daily variations of AOD from AERONET observations (OBS) and the WRF-Chem model in different cases (DYN, STA, (a) DYN1, (b) DYN2, (c) DYN3, (d) DYN4, (e) DYN5) during the in March 2020 at four sites (AOE_Baotou, Beijing_RADI, Hetian, Delingha).**

[Figure]

**Figure 7: Regional average of the dust emission flux (blue bar graph, units: μg m⁻² s⁻¹), dust loading (red dots, units: mg m−2), and dust deposition flux (gray dots, units: μg m⁻² s⁻¹) in the study area (13°N–51°N and 78°E–127°E), Taklimakan Desert (36°N–43°N and 78°E–94°E), and Gobi Desert (38°N–46°N and 96°E–110°E) in March 2020 in the different cases (DYN, DYN1, DYN2, DYN3, DYN4, and DYN5).**

[Figure]

**Figure 8: Spatial distribution of dust emission (color contour, units: μg m⁻² s⁻¹) from the WRF-Chem simulations in March 2020 in (a) case DYN and the difference between the DYN case and the DYN1, DYN2, DYN3, DYN4, and DYN5 cases: (b) DYN1−DYN, (c)**

[Figure]

**Figure 9: The same as Figure 8 but for dust loading (units: mg m$^{-2}$).**

[Figure]

**Figure 10: Profile of dust concentration at 40°N in March 2020 in the different DYN cases (color contour, units: μg m$^{-3}$; vector, consisting of vertical velocity in units of 10 cm s$^{-1}$ and zonal wind in m s$^{-1}$). (a) Dust concentration profile for case DYN. (b), (c), (d), (e), and (f) represent the difference between the DYN case and the DY1, DYN2, DYN3, DYN4, and DYN5 cases (DYN1−DYN, DYN2−DYN, DYN3−DYN, DYN4−DYN, and DYN5−DYN), respectively.**

[Figure]

**Figure 11: The same as Figure 8 but for the dust dry deposition flux (color contour, units: μg m⁻² s⁻¹).**

[Figure]

**Figure 12: The same as Figure 8 but for dust radiative forcing at the top of the atmosphere (units: W m⁻²).**

We also updated the spatial distribution of surface bareness and topographic characteristics to the East Asia in the revised manuscript.

[Figure]

**Figure 1: Spatial distribution of the global surface bareness in March 2020. (a) thr is characterized by 0.12. (b) Surface bareness difference between thr=0.15 and thr=0.12. (c) Surface bareness difference between thr=0.17 and thr=0.12.**

[Figure]

**Figure 2: Spatial distribution of global topographic features. (a) Calculation resolution of 10°×10°. (b) Calculation resolution of 15°×15°.**

**Reference:**

Chen, S., Zhao, C., Qian, Y., Leung, L. R., Huang, J., Huang, Z., Bi, J., Zhang, W., Shi, J., Yang, L., Li, D., and Li, J.: Regional modeling of dust mass balance and radiative forcing over East Asia using WRF-Chem, Aeolian Res., 15, 15–30, doi: 10.1016/j.aeolia.2014.02.001, 2014.

Gong, S. L., Zhang, X. Y., Zhang, T. L., McKendry, I. G., Jaffe, D. A., and Lu, N. M.: Characterization of soil dust aerosol in China and its transport and distribution during 2001 ACE-Asia:2. model simulation and validation, J. Geophys. Res.-Atmos., 108 (D9), https://doi.org/10.1029/2002JD002633, 2003.

Huneeus, N., Schulz, M., Balkanski, Y., Griesfeller, J.,Prospero, J., Kinne, S., Bauer, S., Boucher, O., Chin, M., Dentener, F., Diehl, T., Easter, R., Fillmore, D., Ghan, S., Ginoux, P., Grini, A., Horowitz, L., Koch, D., Krol, M. C., Landing, W., Liu, X., Mahowald, N., Miller, R., Morcrette, J. J., Myhre, G., Penner, J.,Perlwitz, J., Stier, P., Takemura, T., Zender. C. S.: Global dust model intercomparison in AeroCom phase i. Atmos. Chem. Phys., 11 (15), 7781–7816, https://doi.org/10.5194/acp-11-7781-2011, 2011.

Huang, Z. W., Huang, J. P., Bi, J. R., Wang, G. Y., Wang, W. C., Fu, Q., Li, Z. Q., Tsay, S. C., and Shi, J. S.: Dust aerosol vertical structure measurements using three MPL lidars during 2008 China-U.S. joint dust field experiment, J. Geophys. Res.-Atmos., 115, D00K15, https://doi.org/10.1029/2009JD013273, 2010.

Kim, D., Chin, M., Bian, H., Tan, Q., et al.: The effect of the dynamic surface bareness on dust source function, emission, and distribution. J. Geophys. Res.-Atmos., 118 (2), 871–886, https://doi.org/10.1029/2012JD017907, 2013.

Kok, J. F., Adebiyi, A. A., Albani, S., Balkanski, Y., Checa-Garcia, R., Chin, M., Colarco, P. R., Hamilton, D. S., Huang, Y., Ito, A., Klose, M., Li, L., Mahowald, N. M., Miller, R. L., Obiso, V., Pérez García-Pando, C., Rocha-Lima, A., and Wan, J. S.: Contribution of the world's main dust source regions to the global cycle of desert dust, Atmos. Chem. Phys., 21, 8169–8193,

https://doi.org/10.5194/acp-21-8169-2021, 2021.

LeGrand, S. L., Polashenski, C., Letcher, T. W., Creighton, G. A., Peckham, S. E., and Cetola, J. D.: The AFWA dust emission scheme for the GOCART aerosol model in WRF-Chem v3.8.1, Geosci. Model Dev., 12, 131–166, https://doi.org/10.5194/gmd-12-131-2019, 2019.

Prospero, J. M., Ginoux, P., Torres, O., Nicholson, S. E., and Gill, T. E.: Environmental characterization of global sources of atmospheric soil dust identified with the NIMBUS 7 Total Ozone Mapping Spectrometer (TOMS) absorbing aerosol product, Rev. Geophys., 40(1), 1002, https://doi.org/10.1029/2000RG000095, 2002.

Shao, Y.P.: A model for mineral dust emission, J. Geophys.Res.-Atmos, 106 (D17), 20239–20254, https://doi.org/10.1029/2001JD900171, 2001.

Shao, Y. P., Klose, M., Wyrwoll, K., H.: Recent global dust trend and connections to climate forcing. J. Geophys. Res.-Atmos., 118 (19), 11,107-11,118, https://doi.org/10.1002/jgrd.50836, 2013.

Tegen, I., Harrison, S. P., Kohfeld, K., Prentice, I. C., Coe, M., and Heimann, M.: Impact of vegetation and preferential source areas on global dust aerosol: Results from a model study, J. Geophys Res.-Atmos., 107 (D21), 4576, https://doi.org/10.1029/2001JD000963, 2002.

Tian, R., Ma, X., and Zhao, J.: A revised mineral dust emission scheme in GEOS-Chem: improvements in dust simulations over China, Atmos. Chem. Phys., 21, 4319－4337, https://doi.org/10.5194/acp-21-4319-2021, 2021.

Uno, I., Wang, Z., Chiba, M., Chun, Y.S., Gong, S.L., Hara, Y., Jung, E., Lee, S.S., Liu, M., Mikami, M., Music, S., Nichovic, S., Satake, S., Shao, Y., Song, Z., Sugimoto, N., Tanaka, T., Westphal, D.L.: Dust model intercomparison (DMIP) study over Asia: Overview. J. Geophys. Res.-Atmos., 111 (D12), D12213, https://doi.org/10.1029/2005JD006575, 2006.

Uno, I., Yumimoto, K., Shimizu, A., Hara, Y., Sugimoto, N., Wang, Z., Liu, Z., and Winker, D. M.: 3D structure of Asian dust transport revealed by CALIPSO lidar and a 4DVAR dust model, Geophys. Res. Lett., 35 (6), 341–356. https://doi.org/10.1029/2007GL032329, 2008.

Zhao, C., Liu, X., Leung, L. R., Johnson, B., McFarlane, S. A., Gustafson Jr., W. I., Fast, J. D., and Easter, R.: The spatial distribution of mineral dust and its shortwave radiative forcing over North Africa: modeling sensitivities to dust emissions and aerosol size treatments, Atmos. Chem. Phys., 10, 8821–8838, https://doi.org/10.5194/acp-10-8821-2010, 2010.

*Referee Comment 1:*

*Chen et al. present a vegetation-dependent – and hence dynamic – dust source function to use with the GOCART dust emission scheme in WRF-Chem. Introducing these dynamic dust sources, the authors aim to address a supposed long-standing neglect of variations in surface bareness in dust modeling (Abstract, lines 16-17). While the subject of the manuscript is highly relevant and there are still important unknowns concerning the representation of dust sources and dust emission in models, I unfortunately do not see much novelty in the presented manuscript for the reasons detailed in the following.*

**Reply:**Thanks for your suggestions. Uncertainties of dust emission simulation can emanate from several sources including wind (Chen et al., 2014), environmental forcing condition patterns (Shao et al., 2013), soil texture (Tian et al., 2021), terrain (Prospero et al., 2002), vegetation characterization (Tegen et al., 2002), poorly or overly tuned empirical constants (Zhao et al., 2010), etc. Researchers have made a lot of progress in improving dust simulation, especially taking into account the vegetation factors in the dust emission simulation improvement. Tegen et al., (2002) used a monthly FPAR (the fraction of absorbed photosynthetically active radiation) limit of 0.25 for grass and annual mean FPAR (0.5) for shrubs. They took detailed consideration of different land cover, different soil textures and particle size distribution. Zender et al., (2003) used a satellite derived vegetation dataset and set a dust emission suppression threshold for 0.3 $m^2$ $m^{-2}$. Klose et al., (2021) mainly concentrated on the effect of vegetation on dust emission through its influence on aerodynamic roughness length. They presented a methodology to account for the wind drag partition effect due to nonerodible roughness elements including vegetation and rocks that protect the bare soil by absorbing part of the surface wind stress (Leung et al., 2023). Their work either consider vegetation as a separate factor in the dust emission flux (Tegen et al., 2002; Zender et al., 2003), or explore the effect of vegetation on dust by considering its influence on drag force (Leung et al., 2023) and atmospheric roughness (Klose et al., 2021). However, due to the strong spatial heterogeneity of global dust source, the accurate description of dust sources in numerical models has also become an important factor affecting the dust simulation uncertainties. To date, few studies have attempted to improve the dust simulation directly focus on the dust source (Kim et al., 2013; 2017).

Research related to East Asian dust source is particularly crucial for the dust numerical simulation in this region. The greening phenomenon, together with global warming and anthropogenic activies, respresents highly credible evidence for global dust source change (Chen et al., 2019; Piao et al., 2019; Wang et al., 2023). The complex spatial distribution of dust sources over East Asia also brings great challenges to dust simulation. Althrough Kim et al. (2013) proposed work on dynamic dust source, their work mainly focused on global dust source variability. East Asia, the second dust source in the world, has significant influence on the global dust cycle and the radiative budget of the Earth-air system (Shao et al., 2011, Wu t al., 2020, Yin et al., 2021). In recent years, due to the "grain-for-green", grazing exclusion practices and climate change (Wang et al., 2023), China has taken a leading position in

greening the world (Chen et al., 2019), accounting for 25% of the global net increase in leaf area. They found that the eastern edge of the Gobi Desert and the northern part of the Taklimakan Desert have experienced significant greening. Moreover, the East Asian topography is complicate. The current dust simulation over East Asia is inaccurate.

Above all, this study aims to improve the dust simulation over East Asia. Based on the uncertainties analysis, the influence of dust source over East Asia is further explored for future research, which impact great on the dust simulation over East Asia. In order to show the novelty of this research more clearly, a detailed description of the main work are displayed in the manuscript in Lines 77–95 as follows:"Research on the impact of vegetation change and topographic characteristics on East Asian dust source change therefore is urgent. In recent years, due to the "grain-for-green", grazing exclusion practices and climate change (Wang et al., 2023), China has taken a leading role in greening the world. Notably, satellite remote sensing has even captured significant greening in the eastern Gobi Desert and the northern Taklimakan Desert (Chen et al., 2019). Time-varying vegetation, an important factor closely associated with dust emission in the dust source regions (Engelstaedter et al., 2003; Zender and Kwon, 2005), was characterized to show the dynamic changes of dust sources in the GOCART dust emission scheme by Kim et al. (2013) for the first time. In addition, the complex topography in East Asia brings great challenges to dust cycle simulation. However, as the main two hotpots in dust study, the dust emission flux from Taklimakan Desert and Gobi Desert differ immensely among different models (Uno et al., 2006), which indicates the importance of accurate updated land use information for models improvement over these dust regions.

Six sensitive experiments are designed in this study to reveal the influence of surface bareness and topographic characteristics on East Asian dust source function. The detailed organization of the paper is as follows. Section 2 describes the construction of the surface bareness map and topographic feature function dataset. The WRF-Chem model, GOCAT dust emission scheme, six sensitivity experiments, and model evaluation data sets used in this study are introduced in detail. Section 3 presents the model evaluation and uncertainty analyses. Section 4 contains the summary and discussion."

*The claim that dust models usually neglect variability in surface bareness is not correct. Dust models have been considering variations in surface bareness, particularly due to vegetation coverage, for a long time, e.g. Tegen et al. (2002), Zender et al. (2003), …, Klose et al. (2021), Leung et al. (2023). However, the influence of dynamic vegetation is not necessarily implemented in a preferential dust source function as done in the present paper, but is used separately either as a multiplicative factor or as a correction function to the threshold friction velocity or friction velocity (drag partitioning). If vegetation is treated that way, a static dust source function is indeed sufficient for those schemes that use it. There are also schemes that do not use preferential source functions, but aim to explicitly represent the land-surface properties and their impacts on dust emissions.*

**Reply:** Thanks for your suggestions. We have revised the relative expression about the neglect of surface bareness The complex spatial distribution of dust sources over East Asia brings great challenges to dust simulation. Dust models have been considering variations in surface bareness, particularly due to vegetation coverage. Tegen et al., (2002) used a monthly FPAR(the fraction of absorbed photosynthetically active radiation) limit of 0.25 for grass and annual mean FPAR (0.5) for shrubs. They took detailed consideration of different land cover, different soil textures and particle size distribution. Zender et al., (2003) used a satellite derived vegetation dataset and set a dust emission suppression threshold for 0.3 $m^2$ $m^{-2}$. Klose et al., (2021) mainly concentrated on the effect of vegetation on dust emission through its influence on aerodynamic roughness length. They presented a methodology to account for the wind drag partition effect due to nonerodible roughness elements including vegetation and rocks that protect the bare soil by absorbing part of the surface wind stress (Leung et al., 2023). Their work either consider vegetation as a separate factor in the dust emission flux (Tegen et al., 2002; Zender et al., 2003), or explore the effect of vegetation on dust by considering its influence on drag force (Leung et al., 2023) and atmospheric roughness (Klose et al., 2021). This study directly focus on the dust source function through the NDVI, which aims to achieve a large change in the dust cycle simulation through simple change.

Moreover, their researches mainly focused on the global dust emission simulation (Tegen et al., 2002; Zender et al., 2003; Klose et al., 2021; Leung et al., 2023). East Asia, the second dust source in the world, has significant influence on the global dust cycle and the radiative budget of the Earth-air system (Shao et al., 2011, Wu t al., 2020, Yin et al., 2021). In recent years, due to the "grain-for-green", grazing exclusion practices and climate change (Wang et al., 2023), China has taken a leading position in greening the world (Chen et al., 2019), accounting for 25% of the global net increase in leaf area. They found that the eastern edge of the Gobi Desert and the northern part of the Taklimakan Desert have experienced significant greening. Research on the impact of vegetation change on East Asian dust source change therefore is urgent. Moreover, the East Asian topography is complicate. The complex spatial distribution of dust sources over East Asia brings great challenges to dust simulation. With uncertainty analysis, the influence of dynamic dust source on dust cycle simulation is further investigated.

As the complex physical processes involved in dust cycle, the dust simulation in the Taklimakan Desert and the Gobi Desert differ immensely among different models. The simulation uncertainties can emanate from several sources (e.g., erroneous wind or environmental forcing condition patterns, erroneous soil/terrain/vegetation characterization, poorly or overly tuned empirical constants, etc.). The traditional models mainly employ the static dust source without consideration of the effect of the dynamic change of dust source on dust emission, so does the original GOCART dust emission scheme. We explored the effect of changes in vegetation cover and topographic characteristics on dust source function and further impact on dust emission over East Asia, especially Taklimakan Desert and Gobi Desert, with GOCART dust emission scheme in the WRF-Chem. The results show that it is

necessary to couple the dynamic dust source in the WRF-Chem, while the dust source function difference between the dynamic and static dust source could reach 0.2. Moreover, the surface bareness threshold increase further leads to the dust source function change at the desert edge, while coarse topographic calculation resolution results in dust source function decrease in the center part of the Gobi Desert and the eastern part of Taklimakan Desert.

We updated the description that the current numerical model ignore surface bareness change in Lines 16–17,

"The previous studies always employed static land cover in the numerical models, ignoring dynamic variations of dust source and leading to large uncertainties in the dust simulation."

in Lines 54–59,

"Uncertainites of dust emission simulation can emanate from several sources including wind (Chen et al., 2014), environmental forcing condition patterns (Shao et al., 2013), soil texture(Tian et al., 2021), terrain (Prospero et al., 2002), vegetation characterization (Tegen et al., 2002, ), poorly or overly tuned empirical constants (Zhao et al., 2010), etc. However, due to the strong spatial heterogeneity of global dust source (Kok et al., 2021), the accurate description of dust sources in numerical models has also become an important factor affecting the dust simulation uncertainties (Kim et al., 2013)."

in Lines 77–88,

"Research on the impact of vegetation change and topographic characteristics on East Asian dust source change is urgent. In recent years, due to the "grain-for-green", grazing exclusion practices and climate change (Wang et al., 2023), China has taken a leading role in greening the world. The satellite remote sensing has even captured significant greening in the eastern Gobi Desert and the northern Taklimakan Desert (Chen et al., 2019). Time-varying vegetation, an important factor closely associated with dust emission in the dust source regions (Engelstaedter et al., 2003; Zender and Kwon, 2005), was characterized to show the dynamic changes of dust sources in the GOCART dust emission scheme by Kim et al. (2013) for the first time. In addition, the complex topography in East Asia brings great challenges to dust cycle simulation. However, as the main two hotpots in dust study, the dust emission flux from Taklimakan Desert and Gobi Desert differ immensely among different models (Uno et al., 2006), which indicates the importance of accurate updated land use information for models improvement over these dust regions."

*Specific for use with the GOCART dust emission formulation by Ginoux et al. (2001) – which originally indeed does not consider dynamic vegetation – the new dynamic source functions may still be very useful. However, the dynamic source functions presented here are in fact not new, but have already been presented by Kim et al. (2013, 2017).*

**Reply:** Thanks for your suggestions. However, the greening phenomenon, together with global warming and anthropogenic activies, respresents highly credible evidence for global dust source change (Chen et al., 2019; Piao et al., 2019; Wang et al., 2023).

The complex spatial distribution of dust sources over East Asia also brings great challenges to dust simulation, and the present researches about dust are infrequent. Althrough Kim et al. (2013) proposed work on dynamic dust source, their work mainly focused on global dust source variability. East Asia, the second dust source in the world, has significant influence on the global dust cycle and the radiative budget of the Earth-air system (Shao et al., 2011, Wu t al., 2020, Yin et al., 2021). In recent years, due to the "grain-for-green", grazing exclusion practices and climate change (Wang et al., 2023), China has taken a leading position in greening the world (Chen et al., 2019), accounting for 25% of the global net increase in leaf area. They found that the eastern edge of the Gobi Desert and the northern part of the Taklimakan Desert have experienced significant greening. Research on the impact of vegetation change on East Asian dust source change therefore is urgent. Moreover, the East Asian topography is complicate. The current dust simulation over East Asia is inaccurate. Therefore, considering the natural and anthropogenic factors, we give six East Asian dust sources for better dust simulation.

This study mainly focused on the impact of dynamic dust sources on East Asian dust simulation. East Asia, the second dust source in the world, has significant influence on the global dust cycle and the radiative budget of the Earth-air system (Shao et al., 2011, Wu et al., 2020, Yin et al., 2021). In recent years, due to the "grain-for-green", grazing exclusion practices and climate change (Wang et al., 2023), China has taken a leading position in greening the world (Chen et al., 2019), accounting for 25% of the global net increase in leaf area. They found that the eastern edge of the Gobi Desert and the northern part of the Taklimakan Desert have experienced significant greening. Research on the impact of vegetation change on East Asian dust source change therefore is urgent. Moreover, the East Asian topography is complicate. The complex spatial distribution of dust sources over East Asia brings great challenges to dust simulation. With uncertainty analysis, the influence of dynamic dust source on dust cycle simulation is further investigated. Specifically, six sensitive experiments are designed in this study to reveal the influence of surface bareness and topographic characteristics calculation resolution on East Asian dust source function. By using fine NDVI data, a dynamic dust source data set with higher resolution than previous studies was constructed to explore the influence of vegetation change on dust source function. This study impact great on the dust numerical simulation over East Asia.

Further improvements to the dynamic dust sources over East Asia have been carried out in this study based on their work (Kim et al., 2013). We further investigate the influence of surface bareness and topographic characteristics on East Asian dust sources. Kim et al., (2013) constructed a group of dynamic dust source (NDVI threshold: 0.15, topographic calculation resolution: 10°×10°) to prove the superiority of dynamic dust source. Six dynamic dust source experiments (NDVI threshold: 0.12, 0.15, 0.17; topographic calculation resolution: 10°×10°, 15°×15°) were carried out to reveal the effect of surface bareness and topographic characteristics on dynamic dust sources and dust cycle simulation in this study. Second, their study used a coarser spatial resolution in NDVI (1° × 1°), while that in this study is (grid number: 7200

(west-east) × 3600 (south-north)). The spatial distribution of dynamic dust source in their study is 1° × 1°, while that in this study is 0.05° × 0.05°. Third, 16 kinds of land covers types are taken into consideration of dynamic dust source construction, while that in Kim et al. (2013) is 12. Moreover, this study mainly aims to explore the influence of surface bareness and calculated grid resolution on dynamic dust source.

Due to the complexity of the terrain in East Asia and the vegetation change in recent years, this study provides scientific references for future researchers in dust emission, dust deposition, dust transportation, dust radiative effects and so on. Therefore, we made a further extension based on the research of Kim et al. (2013). We also further emphasize the importance of this study in the introduction part in Lines 526–540 as follows: "Recently, in the influence of East Asian greening, this study is particularly important to explore the impact of NDVI threshold on the dust source change in East Asia. Kim et al., (2013) provide excellent insights on the dust simulation from GOCART dust emission scheme. Based on the dynamic dust source constructed by Kim et al.(2013), this study further improved the dynamic dust source in East Asia to more appropriately explore the influence of surface bareness and topographic characteristics on the dust source in East Asia. Specifically, Kim et al., (2013) constructed a group of dynamic dust source (NDVI threshold: 0.15, topographic calculation resolution: 10°×10°) to prove the superiority of dynamic dust source. Six dynamic dust source experiments ((NDVI threshold: 0.12, 0.15, 0.17; topographic calculation resolution: 10°×10°, 15°×15°) were carried out to reveal the effect of surface bareness and topographic characteristics on dynamic dust sources and dust cycle simulation in this study. Second, their study used a coarser spatial resolution in NDVI (1° × 1°), while that in this study is (grid number: 7200 (west-east) × 3600 (south-north)). The spatial distribution of dynamic dust source in their study is 1° × 1°, while that in this study is 0.05° × 0.05°. Third, 16 kinds of land covers types are taken into consideration of dynamic dust source construction, while that in Kim et al. (2013) is 12."

*Applying the dynamic source functions, the authors then present sensitivity experiments investigating the impact of those functions on the dust cycle in East Asia. Unfortunately, the discussion of results of this part remains very descriptive and does not go into enough detail to provide new insights.*

**Reply:** Thanks for your suggestions. We have added more detailed discussion on the research results and uncertainties in this study in Lines 502–509 as follows: "The dust source function of the Taklimakan Desert and Gobi Desert also decrease at an annual rate of $2.14 \times 10^{-4}$ and $3.05 \times 10^{-4}$. The spatial distribution of the dynamic dust source was significantly wider than that of the static dust source. New dust sources appeared in southern Mongolia and the Gobi Desert at the China–Mongolia border, with a dust source of >0.1. The WRF-Chem effectively improved dust simulation across the dust source regions when coupled with dynamic dust sources. The spatial distribution of the AOD simulation over the Taklimakan Desert in the dynamic dust sources cases was consistent with that of MODIS AOD, while that with static dust sources showed more poor performance."

In Lines 511–524 as follows:"This study also examines the uncertainties resulting from dynamic dust sources in dust cycle simulation over East Asia. Six sensitive experiments were carried out to explore the influence of surface bareness and calculated grid resolution on dynamic dust source with six groups of experiments.

Our results revealed that changes in surface bareness and topographic characteristics could change basal parameters of dust (dust AOD, dust emission flux, dust loading, dust concentration at different height layers, and dust dry deposition) by influencing the dynamic dust sources. Overall, the NDVI threshold change mainly increase the dust source function at the edge of Taklimakan Desert and Gobi Desert. The East Asia is characterized with intricate topography. When the calculated resolution is 15°×15°, the dust source function in the central part of dust sources decreased by 0.1 compared that with calculated resolution of 10°×10°. Surface bareness and topographic characteristics considerably affect the spatial distribution and numerical value of the dust cycle. The dust cycle simulation in the different DYN cases differed from each other, but changes in the value and spatial distribution were consistent with the changes in the dynamic dust sources function. The simulation of the dust cycle in the eastern Gobi Desert, Taklimakan Desert, and North China increases as the surface bareness increases, but that in the western Taklimakan Desert and central Gobi Desert decreases with coarse topographic characteristics calculation resolution."

In Lines 526–540 as follows:"Recently, in the influence of East Asian greening, this study is particularly important to explore the impact of NDVI threshold on the dust source change in East Asia. Kim et al., (2013) provide excellent insights on the dust simulation from GOCART dust emission scheme. Based on the dynamic dust source constructed by Kim et al.(2013), this study further improved the dynamic dust source in East Asia to more appropriately explore the influence of surface bareness and topographic characteristics on the dust source in East Asia. Specifically, Kim et al., (2013) constructed a group of dynamic dust source (NDVI threshold: 0.15, topographic calculation resolution: 10°×10°) to prove the superiority of dynamic dust source. Six dynamic dust source experiments ((NDVI threshold: 0.12, 0.15, 0.17; topographic calculation resolution: 10°×10°, 15°×15°) were carried out to reveal the effect of surface bareness and topographic characteristics on dynamic dust sources and dust cycle simulation in this study. Second, their study used a coarser spatial resolution in NDVI (1°×1°), while that in this study is (grid number: 7200 (west-east) × 3600 (south-north)). The spatial distribution of dynamic dust source in their study is 1° × 1°, while that in this study is 0.05° × 0.05°. Third, 16 kinds of land covers types are taken into consideration of dynamic dust source construction, while that in Kim et al. (2013) is 12."

In Lines 545–548 as follows:"We set a uniform bareness threshold threshold for the whole study area. However, it is worthwhile to develop a region-dependent NDVI threshold considering the characteristics of different regions to further explore the dynamic dust source in the future studies."

*From a more technical perspective, very little information is provided about the simulation setup and it is not clear to me how the simulated dust deposition fluxes can be about two orders of magnitude larger than the dust emission fluxes. Those should typically be on the same order of magnitude.*

**Reply:** Thanks for your suggestions. When we calculate the dust deposition flux, it is post-processed in the previous manuscript and the unit of dust deposition is g day$^{-1}$. For less readers' confusion, we have updated it to the same unit ($\mu$g m$^{-2}$ s$^{-1}$).

[Figure]

**Figure 8 Spatial distribution of dust emission in March 2020 (color contour, units: µg m−2 s−1) from the WRF-Chem simulations in (a) case DYN and the difference between the DYN case and the DYN1, DYN2, DYN3, DYN4, and DYN5 cases: (b) DYN1–DYN, (c) DYN2–DYN, (d) DYN3–DYN, (e) DYN4–DYN, and (f) DYN5–DYN.**

[Figure]

**Figure 11 The same as Figure 8 but for the dust dry deposition flux (color contour, units: ug m⁻²s⁻¹).**

*I hope that the authors will keep their motivation to advance dust modeling in the future.*

**Reply:** Thanks for your valuable suggestions. The reviewers provided a lot of valuable opinions on the innovation and the accuracy of this study. The comments on the revised manuscript have been substantially carried out. We have updated all the figures in this study to an updated version of WRF-Chem (version 4.4.2), adding a specific description of the dust module in the WRF-Chem. In addition, we also highlight a series of improvement in this study based on the work of Kim et al. (2013). We sincerely hope to get reviewers' recognition to our work. The significance of this study in East Asia is also emphasized in the introduction sections in the revised manuscript.

Reference:

Chen, C., Park, T., Wang, X. H., Piao, S.L., Xu, B.D., Chaturvedi, R.K., Fuchs, R., Brovkin, V., Ciais, P., Fensholt, R., Tømmervik, H., Bala, G., Zhu, Z.C., Nemani, R.R., Myneni, R.B. China and India lead in greening of the world through land-use management, Nat Sustain., 2, 122–129. https://doi.org/10.1038/s41893-019-0220-7, 2019.

Chen, S., Zhao, C., Qian, Y., Leung, L. R., Huang, J., Huang, Z., Bi, J., Zhang, W., Shi, J., Yang, L., Li, D., and Li, J.: Regional modeling of dust mass balance and radiative forcing over East Asia using WRF-Chem, Aeolian Res., 15, 15–30, doi: 10.1016/j.aeolia.2014.02.001, 2014.

Engelstaedter, S., Kohfeld, K.E., Tegen, I., and Harrison, S.P., Controls of dust emissions by vegetation and topographic depressions:An evaluation using dust storm frequency data, Geophys. Res. Lett., 30 (6), 1294, https://doi.org/10.1029/2002GL016471, 2003.

Kim, D., Chin, M., Bian, H., Tan, Q., et al.: The effect of the dynamic surface bareness on dust source function, emission, and distribution.: J. Geophys. Res.-Atmos., 118 (2), 871–886, https://doi.org/10.1029/2012JD017907, 2013.

Kim, D., Chin, M., Kemp, E. M., Tao, Z., Peters-Lidard, C. D., and Ginoux, P.: Development of High-Resolution Dynamic Dust Source Function - A Case Study with a strong Dust Storm in a Regional Model, Atmos. Environ. 159, 11–25, https://doi.org/10.1016/j.atmosenv.2017.03.045, 2017.

Klose, M., Jorba, O., Gonçalves Ageitos, M., Escribano, J., Dawson, M. L., Obiso, V., Di Tomaso, E., Basart, S., Montané Pinto, G., Macchia, F., Ginoux, P., Guerschman, J., Prigent, C., Huang, Y., Kok, J. F., Miller, R. L., and Pérez García-Pando, C.: Mineral dust cycle in the Multiscale Online Nonhydrostatic AtmospheRe CHemistry model (MONARCH) Version 2.0, Geosci. Model Dev., 14, 6403–6444, https://doi.org/10.5194/gmd-14-6403-2021, 2021.

Leung, D.M., Kok, J.F., Li, L., Okin, G. S., Prigent, C., Klose, M., García-Pando, C. P., Menut, L., Mahowald, N. M., Lawrence, D. M., Chamecki, M.: A new process-based and scale-aware desert dust emission scheme for global climate models – Part I: Description and evaluation against inverse modeling emissions, Atmos. Chem. Phys., 23, 6487–6523, 10.5194/acp-23-6487-2023, 2023.

Shao, Y. P., Wyrwoll, K. H., Chappell, A., Huang, J. P., Lin, Z.H., McTainsh, G.H., Mikami, M., Tanaka, T.Y., Wang, X.L., Yoon, S.C.: Dust cycle: An emerging core theme in Earth system science. Aeolian Res., 2 (4), 181–204, https://doi.org/10.1016/j.aeolia.2011.02.001, 2011.

Shao, Y. P., Klose, M., Wyrwoll, K., H.: Recent global dust trend and connections to climate forcing. J. Geophys. Res.-Atmos., 118 (19), 11,107-11,118, https://doi.org/10.1002/jgrd.50836, 2013.

Uno, I., Wang, Z., Chiba, M., Chun, Y.S., Gong, S.L., Hara, Y., Jung, E., Lee, S.S., Liu, M., Mikami, M., Music, S., Nichovic, S., Satake, S., Shao, Y., Song, Z., Sugimoto, N., Tanaka, T., Westphal, D.L.: Dust model intercomparison (DMIP) study over Asia: Overview. J. Geophys. Res.-Atmos., 111 (D12), D12213,

https://doi.org/10.1029/2005JD006575, 2006.

Wang, X.M., Ge, Q.S., Geng, X., Wang, Z.S., Gao, L., et al.: Unintended consequences of combating desertification in China, Nat Commun., 14, 1139, https://doi.org/10.1038/s41467-023-36835-z, 2023.

Wu, C., Lin, Z., and Liu, X.: The global dust cycle and uncertainty in CMIP5 (Coupled Model Intercomparison Project phase 5) models, Atmos. Chem. Phys., 20, 10401–10425, https://doi.org/10.5194/acp-20-10401-2020, 2020.

Yin, Z. C., Wan Y, Zhang, Y. J. and Wang, H. J.: Why super sandstorm 2021 in North China? Natl Sci. Rev., 9 (3), nwab165. https://doi.org/10.1093/nsr/nwab165, 2021.

Piao, S., Wang, X., Park, T. et al. Characteristics, drivers and feedbacks of global greening. Nat Rev Earth Environ 1, 14 – 27, https://doi.org/10.1038/s43017-019-0001-x, 2020.

Prospero, J. M., Ginoux, P., Torres, O., Nicholson, S. E., and Gill, T. E.: Environmental characterization of global sources of atmospheric soil dust identified with the NIMBUS 7 Total Ozone Mapping Spectrometer (TOMS) absorbing aerosol product, Rev. Geophys., 40(1), 1002, https://doi.org/10.1029/2000RG000095, 2002.

Tegen, I., Harrison, S. P., Kohfeld, K., Prentice, I. C., Coe, M., and Heimann, M.: Impact of vegetation and preferential source areas on global dust aerosol: Results from a model study, J. Geophys Res.-Atmos., 107 (D21), 4576, https://doi.org/10.1029/2001JD000963, 2002.

Tian, R., Ma, X., and Zhao, J.: A revised mineral dust emission scheme in GEOS-Chem: improvements in dust simulations over China, Atmos. Chem. Phys., 21, 4319 – 4337, https://doi.org/10.5194/acp-21-4319-2021, 2021.

Zender, C. S., Bian, H., Newman, D.: Mineral Dust Entrainment and Deposition (DEAD) model: Description and 1990s dust climatology, J. Geophys Res.-Atmos., 108 (D14), 4416. https://doi.org/10.1029/2002JD002775, 2003.

Zender, C. S., and Kwon, E. Y.: Regional contrasts in dust emission responses to climate, J. Geophys. Res.-Atmos., 110, D13201, https://doi.org/10.1029/2004JD005501, 2005.

Zhao, C., Liu, X., Leung, L. R., Johnson, B., McFarlane, S. A., Gustafson Jr., W. I., Fast, J. D., and Easter, R.: The spatial distribution of mineral dust and its shortwave radiative forcing over North Africa: modeling sensitivities to dust emissions and aerosol size treatments, Atmos. Chem. Phys., 10, 8821–8838, https://doi.org/10.5194/acp-10-8821-2010, 2010.

*Referee Comment 2:*

*Review Comments for the manuscript "Impacts of dynamic dust sources coupled with WRF-Chem 3.9.1 on the dust simulation over East Asia" by Chen et al.*

*The authors attempt to improve the dust emissions and transport capability of WRF-Chem for East Asia, by changing the characterization of dust sources. This is accomplished by using an NDVI dataset to estimate surface bareness, basically the way aridity is represented in the dust emission scheme.*

*I have many comments and concerns with the manuscript. My first concern is that the "dynamic dust source" that the authors refer to, is not substantiated in the manuscript. The dynamic nature of a model input can be temporal or spatial, or hopefully both. If the main advantage is the monthly variation of bareness from NDVI (lines 101-102), since the WRF-Chem simulations are essentially for one month only (March 2020), it is impossible to assess how this addition improved dust prediction and also can be named "dynamic dust source".*

**Reply:** We constrcuted monthly dust sources over East Asia in recent 21 years (2001-2021), which could reflect the temporal variations of East Asian dust sources with different land cover constraints. Specifically, results show that dynamic dust sources have pronounced fluctuations in different periods (the revised Fig. 3). Dust eruption occurred frequently in spring over East Asia (Chen et al., 2023), the dust source function of the two deserts are generally larger than 0.3 in March (the revised Fig. 3a). Compared that in July, the dust source function in March is also larger and expand to the edge of the desert (the revised Fig. 3b). Exuberant vegetation is accompanied with low-bareness surface, and the dust source function in July is lower than that in March. The dust source function difference over the Taklimakan Desert and Gobi Desert also peak at 0.21 and 0.19 (the revised Fig. 3b), respectively, which indirectly indicates the seasonal change impact great on the dust source function over East Asia. Moreover, the monthly variation of dust source function reaches the trough value in summer in different cases (the revised Fig. 3c and d). After January and February, the dust source function decrease in March, April and May, which is related with the unfavorable growth of vegetation and large surface bareness in winter. The dynamic dust source function also shows sufficient annual variation characteristics (the revised Fig. 3e and f), changing with the trend of $-2.14 \times 10^{-4}$ ($-3.05 \times 10^{-4}$) for Taklimakan Desert. Notably, the decreasing trend in the Gobi Desert is more immense than that in the Taklimakan Desert.

[Figure]

**Figure 3** Spatial distribution of averaged dust source function in the control experiment (DYN) in (a) March, and the (b) difference of dust source function between March and July from 2001 to 2021. The blue boxes indicate the Taklimakan Desert and the Gobi Desert. Monthly averaged dust source function in different cases from 2001 to 2021 in (c) Taklimakan Desert (36 °N–43 °N and 78 °E–94 °E) and (d) Gobi Desert (38 °N–46 °N and 96 °E–110 °E). Annual variation of dust source function in different cases in (e) Taklimakan Desert and (f) Gobi Desert; shading indicates one standard deviations from the 2001 to 2021 mean.

As an important permanent desert over East Asia, the dust source function of the Taklimakan Desert is larger than that of the Gobi Desert (the revised Fig. 3c and d). The annual variation range of the dust source function in the Taklimakan Desert is 0.14~0.17, while that over the Gobi Desert is wider (0.1–0.158). The dust source function over the two deserts enhance with the surface bareness threshold. When the topographic characteristics calculation resolution changed to 15°×15°, the fluctuations range of dust source function over the Taklimakan Desert is around 0.012, while that in the Gobi Desert is 0.022. Notably, surface bareness and topographic characteristics calculation resolution impact greater on dust source function over the Gobi Desert than that over the Taklimakan Desert. In addition, due to the climatic factors and the implementation of afforestation policy in China in recent years (Wu et al., 2022; Wang et al., 2023), the dust occurrence frequency has decreased, and the dust source function value also show a downward trend. It decrease at a rate of $2.14 \times 10^{-4}$ per year over the Taklimakan Desert and $3.05 \times 10^{-4}$ per year over the Gobi Desert.

Moreover, we chose the configuration of dynamic dust sources similar to those in the work of Kim et al. (2013) to further verify the dynamic nature of dust sources. The dust emission flux from the Gobi Desert was greater than that of the Taklimakan Desert from March 2017 to March 2021, which peaked at 3.78 $\mu g\ m^{-2}\ s^{-1}$ in March 2018. The dust loading in the Taklimakan Desert was slightly lower than that in the Gobi Desert in March 2018 and show a similar pattern in March 2021. In the other years, the dust loading in the Gobi Desert all is greater than that in the Taklimakan Desert. Moreover, the dynamic nature is also reflected in the dust deposition.

[Figure]

**Figure Regional average of the dust emission flux (units: $\mu g\ m^{-2}\ s^{-1}$), dust loading (units: $mg\ m^{-2}$), and dust deposition flux (units: $\mu g\ m^{-2}\ s^{-1}$) in the Taklimakan Desert (orange bar graph, 36°N–43°N and 78°E–94°E), and Gobi Desert (blue bar graph, 38°N–46°N**

**and 96°E–110°E) in different period (March 2017, March 2018, March 2019, March 2020, March 2021) in case DYN2.**

*Second, dust modeling requires a delicate description of how dust particles move horizontally (saltation) and vertically (sandblasting, entrainment, disintegration) and stay in the air, their origin (soil texture, and particle size classification), the dust particle size distribution during atmospheric transport, and, of course, atmospheric conditions. Most atmospheric models (global or regional) that simulate the dust cycle, use some characterization of the aridity of the area that changes temporally and spatially. I don't see how this work can be considered model development, which is the core mission of GMD. The manuscript is mostly representing sensitivity simulations with WRF-Chem, by changing one input parameter that affects dust. If I have misunderstood the authors work, I argue that they should be more explicit on the contribution of their work towards model development.*

**Reply:** Thanks for your valuable suggestions. Saltation refers to a layer of soil moving with wind just above the surface. Three processes are responsible for the entrainment of atmospheric dust particles: aerodynamic lift, saltation bombardment and particle disaggregation. Most of the dust emission schemes need to calculate the saltation based on wind speed and then further convert it to the dust emission. Shao (2001) proposed a dust emission scheme, which takes into account three dust processed including entrainment, saltation bombardment and disintegration. GOCART AFWA dust emission scheme is based on the Marticorena-Bergametti (MB) dust emission scheme. It mainly include two processes, which wind shear triggers the saltation of large particle and saltation bombardment influence the fine-particle emission. However, the current dust emission schemes could be divided into three categories: empirical dust emission scheme, simplified-physical-processed dust emission scheme and detailed-microphysical-process dust emission. The GOCART dust emission scheme is a representative of empirical dust emission scheme, which mainly considers the statistical relationship between dust emission flux and friction velocity. Dust uplifting to the atmosphere is parameterized in GOCART dust emission scheme assuming that the vertical particle flux is proportional to the horizontal wind flux and representing the direct conversion from wind speed to dust emission (Ginoux et al., 2001). The impact of saltation bombardment on mobilization are internalized in the relationship between wind speed and dust emission.

The original GOCART dust scheme (dust_opt=1), a popular dust emission scheme in the modeling community which does not requires difficult soil or surface characteristics, are employed in this study. It also is a highly empirical and relatively simple dust emission scheme, which need necessary input factors including wind speed, soil moisture, air density and generalized soil characteristics. The erodible soil is consist of sand, silt and clay. Specifically, five ranges of dust bins (bin1: 0.1-1.0 μm, bin2: 1.0-1.8 μm, bin3: 1.8-3.0 μm, bin4: 3.0-6.0 μm, bin5: 6.0-10.0 μm) are used in the GOCART aerosol module.

We also revised it in the manuscript in Lines 160–183 as follows: "The current dust emission schemes could be divided into three categories: empirical dust emission

scheme, simplified-physical-processed dust emission scheme and detailed-microphysical-process dust emission. The GOCART dust emission scheme is a representative of empirical dust emission scheme, which mainly considers the statistical relationship between dust emission flux and friction velocity. Dust uplifting to the atmosphere is parameterized in GOCART dust emission scheme assuming that the vertical particle flux is proportional to the horizontal wind flux and representing the direct conversion from wind speed to dust emission (Ginoux et al., 2001). The impact of saltation bombardment on mobilization are internalized in the relationship between wind speed and dust emission.

The GOCART dust emission scheme (dust_opt=1) in the WRF-Chem need necessary input factors including wind speed, soil moisture, air density and generalized soil characteristics. The erodible soil is consist of sand, silt and clay. Specifically, five ranges of dust bins (bin1: 0.1–1.0 μm, bin2: 1.0–1.8 μm, bin3: 1.8–3.0 μm, bin4: 3.0–6.0 μm, bin5: 6.0–10.0 μm) are used in the GOCART aerosol module. Dust emission flux from GOCART is calculated as follows,

$$F = \begin{cases} CSs_p u_{10m}^2 (u_{10m} - u_t), & u_{10m} > u_t \\ 0, & otherwise \end{cases} \quad (3)$$

where C is the constant of the dust emission factor. S is the dust source function based on the topography and surface parameters, and it is used to limit the dust emission area in the study area. The layer 1 of EROD parameter, provided by the WRF-Chem, is used to parameterize the S in Eq. (1). $s_p$ represents the fraction of dust in each bin of particle size in the dust emission. The default $s_p$ from Eq. (3) are {0.1, 0.25, 0.25, 0.25, 0.25}. Additionally, $u_{10m}$ is the 10 m horizontal wind speed near the surface; $u_t$ indicates the threshold wind speed. It is calculated as

$$u_{*t} = 0.129 \frac{(\frac{\rho_g g D_p}{\rho_a})^{0.5}(1+\frac{0.006}{\rho_p g D_p^{2.5}})^{0.5}}{[1.928(a(D_p)^x+0.38)^{0.092}-1]^{0.5}} \quad (4)$$

where x equals 1.56, a equals 1331 cm$^{-x}$."

We also emphasized our contribution for model development in Lines 77–88 as follows:"Research on the impact of vegetation change and topographic characteristics on East Asian dust source change therefore is urgent. In recent years, due to the "grain-for-green", grazing exclusion practices and climate change (Wang et al., 2023), China has taken a leading role in greening the world. The satellite remote sensing has even captured significant greening in the eastern Gobi Desert and the northern Taklimakan Desert (Chen et al., 2019). Time-varying vegetation, an important factor closely associated with dust emission in the dust source regions (Engelstaedter et al., 2003; Zender and Kwon, 2005), was characterized to show the dynamic changes of dust sources in the GOCART dust emission scheme by Kim et al. (2013) for the first time. In addition, the complex topography in East Asia brings great challenges to dust cycle simulation. However, as the main two hotpots in dust study, the dust emission flux from Taklimakan Desert and Gobi Desert differ immensely

among different models (Uno et al., 2006), which indicates the importance of accurate updated land use information for models improvement over these dust regions."

***Third, the manuscript lacks details on the dust emission scheme, specifically how the source function S is calculated, how are sp and ut estimated. My guess is that ut is the threshold friction velocity which is parameterized somehow. All these components must be clearly described in the text, to allow the reader to understand how the authors' addition influences the dust emission scheme.***

**Reply:** The dust source function is calculated by surface bareness and topographic characteristics (Kim et al., 2013, 2017). $s_p$ represents the fraction of dust in each bin of dust particle in the dust emission. The default $s_p$ from Eq. (3) are {0.1, 0.25, 0.25, 0.25, 0.25}. $u_t$ indicates the wind speed threshold at 10 m required for initiating erosion. It is calculated as

$$u_t = 0.129 \frac{(\frac{\rho_g g D_p}{\rho_a})^{0.5}(1 + \frac{0.006}{\rho_p g D_p^{2.5}})^{0.5}}{\left[1.928(a(D_p)^x + 0.38)^{0.092} - 1\right]^{0.5}}$$

where x equals 1.56, a equals 1331 cm$^{-x}$.

We also added more information about the detail of dust emission calculation in Lines 160–183 in the revised manuscript as follows: "The current dust emission schemes could be divided into three categories: empirical dust emission scheme, simplified-physical-processed dust emission scheme and detailed-microphysical-process dust emission. The GOCART dust emission scheme is a representative of empirical dust emission scheme, which mainly considers the statistical relationship between dust emission flux and friction velocity. Dust uplifting to the atmosphere is parameterized in GOCART dust emission scheme assuming that the vertical particle flux is proportional to the horizontal wind flux and representing the direct conversion from wind speed to dust emission (Ginoux et al., 2001). The impact of saltation bombardment on mobilization are internalized in the relationship between wind speed and dust emission.

The GOCART dust emission scheme (dust_opt=1) in the WRF-Chem need necessary input factors including wind speed, soil moisture, air density and generalized soil characteristics. The erodible soil is consist of sand, silt and clay. Specifically, five ranges of dust bins (bin1: 0.1–1.0 μm, bin2: 1.0–1.8 μm, bin3: 1.8–3.0 μm, bin4: 3.0–6.0 μm, bin5: 6.0–10.0 μm) are used in the GOCART aerosol module. Dust emission flux from GOCART is calculated as follows,

$$F = \begin{cases} CSs_p u_{10m}^2(u_{10m} - u_t), & u_{10m} > u_t \\ 0, & otherwise \end{cases} \quad (3)$$

where C is the constant of the dust emission factor. S is the dust source function based on the topography and surface parameters, and it is used to limit the dust emission area in the study area. The layer 1 of EROD parameter, provided by the WRF-Chem, is used to parameterize the S in Eq. (1). $s_p$ represents the fraction of

dust in each bin of particle size in the dust emission. The default $s_p$ from Eq. (3) are {0.1, 0.25, 0.25, 0.25, 0.25}. Additionally, $u_{10m}$ is the 10 m horizontal wind speed near the surface; $u_t$ indicates the threshold wind speed. It is calculated as

$$u_{*t} = 0.129 \frac{(\frac{\rho_g g D_p}{\rho_a})^{0.5}(1+\frac{0.006}{\rho_p g D_p^{2.5}})^{0.5}}{[1.928(a(D_p)^x+0.38)^{0.092}-1]^{0.5}} \quad (4)$$

where x equals 1.56, a equals 1331 cm$^{-x}$"

*The evaluation of the dust simulation also lacks robustness. Even though AOD is a very important component, the evaluation must also include dust concentrations or emissions, at least some PM10 measurements that are more readily available. The calculation of AOD depends on how dust is emitted, but there are other aerosol optical characteristic components that dilute a direct evaluation of the dust emission scheme. The same stands for dust deposition.*

**Reply:** Thanks for your suggestions. Two sites over the Taklimakan Desert are chosen in this study. The result show that the dynamic dust sources is better in simulating the PM$_{10}$ peak in the dust event compared that with the static dust source.

[Figure]

**Figure 6 c, d: The daily variation of PM$_{10}$ in Hetian and Delingha in March 2020.**

*Limited area models like WRF-Chem face other constraints, such as the lateral and initial boundary conditions that influence dust production and transport processes. How did those constraints influence the WRF-Chem simulations?*

**Reply:** Thanks for your valuable suggestions. Accurately representing the initial and boundary condition is an important issue for dust numerical simulation (Khan et al., 2019). The Final Operational Global Analysis (FNL) from National Centers for Environmental Prediction (NCEP) and National Center for Atmospheric Research

(NCAR), characterized with the horizontal resolution of 1° × 1° and the time intervals of 6 hours, are employed to generate the initial and lateral boundary condition for the meteorological parameters in this study. The MOZBC utility was used to update chemical initial and lateral boundary conditions of the WRF-Chem using CAM-chem (or. waccm), which is a component of the NCAR Community Earth System Model (CESM) and used for the simulation of global tropospheric and stratospheric atmospheric composition.

We also updated the related information about the lateral and initial boundary condition description in the revised manuscript in Lines 150–157 as follows: "Accurately representing the initial and boundary condition is an important issue for dust numerical simulation (Khan & Kumar, 2019). The Final Operational Global Analysis (FNL) from National Centers for Environmental Prediction (NCEP) and National Center for Atmospheric Research (NCAR), characterized with the horizontal resolution of 1° × 1° and the time intervals of 6 hours, are employed to generate the initial and lateral boundary condition for the meteorological parameters in this study. The MOZBC utility was used to update chemical initial and lateral boundary conditions of the WRF-Chem using CAM-chem, which is a component of the NCAR Community Earth System Model (CESM) and used for the simulation of global tropospheric and stratospheric atmospheric composition."

*In line 150, the authors mention that "GOCART also has been widely welcomed by various numerical models and show excellent performance on dust emission over East Asia (Chen et al., 2014, 2017)." If the performance is excellent, what is the point of this work?*

**Reply:** Thanks for your suggestions. GOCART dust emission scheme is relatively simple and easily portable, which has been widely used in dust emission simulation. However, it is still worth improving on the details of dust simulation calculation. The dust source in the previous GOCART simulation was decided by the 1987 annual averaged satellite land cover data from AVHRR (DeFries and Townshend, 1994), which does not conclude the time variation of dust source. Using the MODIS NDVI data and referring to the previous ideas (Kim et al., 2013, 2017), the constructed dust source function are employed to the GOCART dust emission scheme. The constructed dynamic dust source function in the WRF-Chem could effectively improve dust cycle simulation with a relative simple method. However, different from the previous studies, this study mainly focuses on the dynamic dust source changes over East Asia and aims to explore the influence of surface bareness and topographic characteristics calculation on dynamic dust source.

We also removed the expression of "GOCART also has been widely welcomed by various numerical models and show excellent performance on dust emission over East Asia (Chen et al., 2014, 2017)."

We also revised it in Lines 160–168 as follows: "The current dust emission schemes could be divided into three categories: empirical dust emission scheme, simplified-physical-processed dust emission scheme and detailed-microphysical-process dust emission. The GOCART dust emission scheme is a representative of empirical dust emission scheme, which mainly considers the

statistical relationship between dust emission flux and friction velocity. It has been widely used in dust emission simulation. Dust uplifting to the atmosphere is parameterized in GOCART dust emission scheme assuming that the vertical particle flux is proportional to the horizontal wind flux and representing the direct conversion from wind speed to dust emission (Ginoux et al., 2001). The impact of saltation bombardment on mobilization are internalized in the relationship between wind speed and dust emission."

*Finally, the manuscript needs thorough English grammar editing. There are many instances in the text that the tense is wrong, there is a use of plural instead of singular nouns (e.g. line 128 "WRF-Chem Models") and other grammatical errors.*

**Reply:** Thanks for your suggestions. We have completely finished the English grammar editing for the revised manuscript.

*I believe the manuscript needs extensive revisions to reach the standards of GMD and be considered for publication. I urge the authors to follow the comments and suggestions and improve the quality of the paper.*

**Reply:** Thanks for your valuable suggestions. The comments on the revised manuscript have been substantially carried out. We also revised all figures and main text in the manuscript, We sincerely hope to get reviewers' recognition to our work.

**References:**

Khan, A.W., & Kumar, P.: Impact of chemical initial and lateral boundary conditions on air quality prediction. Adv. Space Res., 64 (6), 1331-1342. https://doi.org/10.1016/j.asr.2019.06.028, 2019.